# Synthetic ground motions in heterogeneous geologies from various sources: the HEMEW$^{\text{S}}$-3D database

Fanny Lehmann[1,2], Filippo Gatti[2], Michaël Bertin[1], and Didier Clouteau[2]

[1]CEA, DAM, DIF, F-91297 Arpajon, France
[2]LMPS - Laboratoire de Mécanique Paris-Saclay, Université Paris-Saclay, CentraleSupélec, ENS Paris-Saclay, CNRS, Gif-sur-Yvette, France

**Correspondence:** Fanny Lehmann (fanny.lehmann@centralesupelec.fr)

**Abstract.** The ever-improving performances of physics-based simulations and the rapid developments of deep learning are offering new perspectives to study earthquake-induced ground motion. Due to the large amount of data required to train deep neural networks, applications have so far been limited to recorded data or two-dimensional simulations. To bridge the gap between deep learning and high-fidelity numerical simulations, this work introduces a new database of physics-based earthquake simulations.

The HEMEW$^{\text{S}}$-3D database comprises $30\,000$ simulations of elastic wave propagation in three-dimensional (3D) geological domains. Each domain is parametrized by a different geological model built from a random arrangement of layers augmented by random fields that represent heterogeneities. Elastic waves originate from a randomly located point-wise source parametrized by a random moment tensor. For each simulation, ground motion is synthetized at the surface by a grid of virtual sensors. The high frequency of waveforms ($f_{max} = 5\,\text{Hz}$) allows extensive analyses of surface ground motion.

Existing and foreseen applications range from statistic analyses of the ground motion variability and machine learning methods on geological models, to deep learning-based predictions of ground motion depending on 3D heterogeneous geologies and source properties. Data are available at https://doi.org/10.57745/LAI6YU (Lehmann, 2023).

## 1 Introduction

Deep learning has a long tradition in seismology thanks to large networks of sensors recording earthquakes worldwide. Applications are extremely diverse, in terms of methods, data, and scientific goals (see e.g. Mousavi and Beroza (2023) for a review). Detecting earthquakes and discriminating them from other events such as explosions, quarry blasts, or seismic noise are the most common applications of deep learning in seismology (Mousavi and Beroza, 2023). A wide variety of methods are also devoted to characterizing earthquakes from ground motion recordings, for instance to estimate source mechanisms, earthquake location, and magnitude. The rapid improvements of deep learning in the last few years have even enabled its use in operational frameworks, thereby providing real-time predictions of earthquake parameters (Zhu et al., 2022).

However, all those methods rely on databases of seismic waveforms. While there exist several curated databases of recorded ground motion, they are sparse in regions with low-to-moderate seismicity or poor instrumental coverage (Bahrampouri et al., 2021; Michelini et al., 2021; Mousavi et al., 2019). In those cases, numerical simulations are a great opportunity to complement

existing databases. Simulations rely on computational schemes to solve the wave propagation equations from the earthquake source to the Earth surface; and provide synthetic waveforms at any spatial point of the simulation domain. Results of 3D physics-based simulations have been compiled for several past earthquakes in e.g. the BB-SPEEDset dataset (Paolucci et al., 2021), the SCEC Broadband platform (Maechling et al., 2015), but the number of simulations is not appropriate for machine learning approaches.

In fact, physics-based simulations show several limitations. Firstly, they require a detailed description of the ground properties that define the physical behaviour of the waves propagating in the Earth. Especially, ground properties should be given as three-dimensional (3D) geological models since 3D features have crucial effects that are not accounted for in two-dimensional (2D) settings (e.g. sedimentary basins leading to site effects) (Moczo et al., 2018; Smerzini et al., 2011; Zhu et al., 2020). Since extensive geophysical investigations are needed to obtain 3D geological models, they are rare, and when existing, they are still limited by epistemic uncertainties. Therefore, when trying to reproduce an earthquake with physics-based numerical simulations, uncertainties can be represented by random heterogeneities added to the reference model to introduce variability (Chaljub et al., 2021; Lehmann et al., 2022).

Quantifying the effects of 3D geological features is made more difficult by the second limitation of physics-based simulations, which is their high computational cost, especially when dealing with high frequencies and large spatial domains. Despite relying on High-Performance Computing (HPC) frameworks, seismic waves propagation simulations can reach tens to hundreds of thousands of equivalent core.hours (Fu et al., 2017; Heinecke et al., 2014; Poursartip et al., 2020). Since computational costs prevent statistical studies on synthetic waveforms due to a limited number of simulations per geological model, deep learning represents a promising alternative to obtain waveforms.

When predicting the surface ground motion generated by an earthquake, it is important to obtain time-series that describe the temporal evolution of shaking, and not only scalar features (such as Peak Ground Acceleration, Cumulative Absolute Velocity) that give useful but limited information. Physics-Informed Neural Networks (PINNs, Raissi et al. (2019)) successfully solved the wave equation (Ding et al., 2023; Karimpouli and Tahmasebi, 2020; Moseley et al., 2020; Rasht-Behesht et al., 2022; Ren et al., 2024; Song et al., 2023; Wu et al., 2023). However, applications are mainly limited to 2D domains and models cannot extrapolate to another geological configuration than the one used in the training phase. Alternatively, generative methods have been used to enhance existing numerical simulations, by increasing their spatial resolution (e.g. Gadylshin et al., 2021) or their frequency content (e.g. Gatti and Clouteau, 2020).

The recent emergence of Scientific Machine Learning (SciML) is offering a new paradigm to predict physics-based ground motion parametrized by 3D ground properties and source parameters, with intrinsic generalization ability to various resolutions and geological configurations. SciML has led to significant scientific developments in communities with large, reliable, and freely available databases. For instance, in numerical weather prediction, Bonev et al. (2023) and Pathak et al. (2022) took advantage of the ERA5 dataset provided by the European Centre for Medium-Range Weather Forecasts (Hersbach et al., 2020). In seismology, Mousavi and Beroza (2023) pointed out that "the limitations on training data and generalization are the main challenges in solving inverse and forward problems using supervised [Deep Neural Networks]."

In this work, we describe the first open database of seismic simulations associated with 3D heterogeneous geological models. The HEMEW$^S$-3D (HEterogeneous Materials and Elastic Waves with Source variability in 3D) database contains $30\,000$ high-fidelity simulations in 3D domains of size $9.6\,\text{km} \times 9.6\,\text{km} \times 9.6\,\text{km}$. This represents a challenging computational task accounting for $9 \times 10^5$ CPU hours and $4.4\,\text{MWh}$ in total. Ground motion was synthetized at the surface of the simulation domain for $8\,\text{s}$ on a grid of $32 \times 32$ virtual sensors. Data are available at https://doi.org/10.57745/LAI6YU (Lehmann, 2023).

In the following, Section 2 provides an overview of existing datasets in related fields, Section 3 describes the geological models, sources, and surface wavefields in the database, Section 4 analyses physical characteristics, Section 5 illustrates applications, and Section 6 discusses limitations and perspectives.

## 2 Related Work

Datasets of recorded ground motion have enabled major deep learning applications in seismology but they have several limitations in data scarce regions. In this section, we focus on datasets with 2D or 3D data used in geophysics and seismology with SciML applications. Due to the mathematical similarities between wave propagation and fluid flow (both are governed by hyperbolic equations), related studies are reviewed beyond the field of seismology. This highlights the challenges of high-fidelity numerical simulations for deep learning applications.

### 2.1 Datasets of ground motion simulations

Ground motion simulations of past earthquakes have been collected in databases for model verification, characterization of complex near-field conditions, and machine learning purposes. The BB-SPEEDset provides 3D simulations of 16 earthquake scenarios in various regions of the globe (Paolucci et al., 2021). The SCEC Broadband Platform also provides simulations of 17 past earthquakes with different source models (Maechling et al., 2015). Other studies focus on specific regions, e.g. the Hayward fault (Petrone et al., 2021) and Turkey (Altindal and Askan, 2022). The limited number of scenarios makes the above-mentioned databases less suitable for machine learning purposes where dataset variability and size are crucial factors. In addition, although providing some variability on the source, those datasets do not consider varying geological models, but focus on regional validated geological models.

### 2.2 3D datasets

Due to the high computational costs of solving 3D Partial Differential Equations (PDEs), only very few 3D datasets are available. CO2 underground storage has been explored with SciML based on 3D numerical simulations (Grady et al., 2023; Wen et al., 2023; Witte et al., 2023). To support the study of Witte et al. (2023), Annon (2022) provided 4,000 simulation results for 3D CO2 flow through geological models based on the Sleipner dataset complemented by random fields (Equinor, 2020). The Kimberlina dataset also contains 6,000 CO2 leakage rates simulations (Mansoor et al., 2020). However, the geological models in both databases are all variants of the geological model carefully estimated for a given region, thereby limiting the reproducibility in other areas.

**Table 1.** Summary of datasets providing geological models and seismic wavefields. Domain: size of the physical domain with the number of grid points given in parenthesis, in (width, depth) for 2D datasets, in (width, length, depth) for 3D datasets. Dimension of seismic wavefields: (receivers along width, time steps) for 2D datasets, (receivers along width, receivers along length, time steps) for 3D datasets. "k" stands for thousand, "m" stands for million.

| Dataset | train/test | Geological models | | | Seismic wavefields | | |
|---|---|---|---|---|---|---|---|
| | | Domain | Values $V_S$ | Construction | Dimensions | Equation | Source |
| Noddyverse (Jessell et al., 2022) | 1m/- | $4 \times 4 \times 4 \mathrm{km}^3$ ($200 \times 200 \times 200$) | categorical | succession of geological events | N/A | N/A | N/A |
| OpenFWI (Deng et al., 2022) | 408k/62k | $0.7 \times 0.7 \mathrm{km}^2$ ($70 \times 70$) | 1500; 4500 m/s | mathematical, from recorded images, and from geological faults | $70 \times 1000$ | 2D acoustic | 5 fixed sources at the surface |
| OpenFWI Kimberlina CO2 (Deng et al., 2022) | 15k/4k | $4 \times 1.4 \mathrm{km}^2$ ($401 \times 141$) | 1200; 3600 m/s | from real data | $101 \times 1251$ | 2D acoustic | 9 fixed sources at the surface |
| OpenFWI Kimberlina 3D (Deng et al., 2022) | 1.6k/163 | $4 \times 4 \times 3.5 \mathrm{km}^3$ ($400 \times 400 \times 350$) | ? | from real data | $40 \times 40 \times 5001$ | 3D acoustic | 25 fixed sources at the surface |
| $\mathbb{E}$**FWI** (Feng et al., 2023) | 144k/24k | $0.35 \times 0.35 \mathrm{km}^2$ ($70 \times 70$) | 612; 3000m/s | mathematical and from geological faults | $70 \times 1000$ | 2D elastic | 5 fixed sources at the surface |
| **HEMEW$^S$-3D** (**this study**) | 27k/3k | $9.6 \times 9.6 \times 9.6 \mathrm{km}^3$ ($32 \times 32 \times 32$) | 1071; 4500 m/s | horizontal layers + random fields | $32 \times 32 \times 800$ | 3D elastic | 1 source with random location and random orientation |

## 2.3 Geophysical datasets

A few datasets of realistic geological units have been developed, such as the Noddyverse dataset of 3D geological models (Jessell et al., 2022). In this dataset, geological models result from the deformation of horizontal layers by successive geological events (folds, faults, unconformities, dykes, plugs, shear zones, and tilts) but no associated ground motion is provided. Along the same line of geological deformation, the OpenFWI database combines geological models with associated waveforms, and targets 2D geophysical inversion as the main application (Deng et al., 2022). OpenFWI contains geological models made of horizontal and non-horizontal layers with various folds. It also includes real geological models from field survey areas and models of $CO_2$ geological storage. To generate the wavefields, the acoustic wave equation is solved in the 2D domains. Waves originate from a line of sources at the surface and wavefields are acquired on a line of receivers at depth. The $\mathbb{E}^{FWI}$ database is an extension of OpenFWI to the elastic wave equation, providing two-component ground motion time series (Feng et al., 2023).

Several other studies have computed simulation outputs for the acoustic or elastic wave equation but data are not public (e.g. Liu et al., 2021; Ovadia et al., 2023; Zhang et al., 2023). Table 1 summarizes the characteristics of the public datasets and shows that no database provides solutions of the elastic wave equation in 3D domains. Our HEMEW$^S$-3D database intends to fill this gap.

## 3 Dataset creation

### 3.1 The elastic wave equation

Elastodynamics describes reversible wave propagation phenomena in solid and fluid domains. In solid mechanics, the solution is represented by a displacement field $\boldsymbol{u} \in \mathbb{R}^3$ propagating in a 3D Euclidean space. We consider a truncated propagation domain $\Omega = [0; L]^3$ with absorbing boundary conditions all around, except the traction-free top surface; and a solution $\boldsymbol{u} : \Omega \times [0, T] \to \mathbb{R}^3$. The domain length is fixed to L $= 9.6\,\mathrm{km}$ and the total time is T $= 8\,\mathrm{s}$. In its most general form, the elastic wave equation writes

$$\rho \frac{\partial^2 \boldsymbol{u}}{\partial t^2} = \nabla \lambda (\nabla \cdot \boldsymbol{u}) + \nabla \mu \left[ \nabla \boldsymbol{u} + (\nabla \boldsymbol{u})^T \right] + (\lambda + 2\mu) \nabla (\nabla \cdot \boldsymbol{u}) - \mu \nabla \times \nabla \times \boldsymbol{u} + \boldsymbol{f} \tag{1}$$

where $\rho : \Omega \to \mathbb{R}$ is the material unit mass density, $\lambda : \Omega \to \mathbb{R}$, $\mu : \Omega \to \mathbb{R}$ are the Lamé parameters, characterizing the thermodynamically reversible mechanical behaviour of the material, and $\boldsymbol{f}$ is the body force distribution. In geomechanics, properties $\rho$, $\lambda$, and $\mu$ are rarely independently characterized due to a lack of measurements. Therefore, it is legitimate to assume that there is a single informative variable from which all parameters can be deduced. In this work, the velocity of shear waves $V_S$ is the informative variable. Equation 1 can then be rewritten under the general form

$$\mathcal{L}(V_S, \boldsymbol{u}) = \boldsymbol{f} \tag{2}$$

## 3.2 Earthquake source

In our database, the forcing term $\boldsymbol{f}(\boldsymbol{x}, t) = \mathrm{div}\,\boldsymbol{m}(\boldsymbol{x}) \cdot s(t)$ is the divergence of a moment tensor density $\boldsymbol{m}$, localized at a point-wise location. $\boldsymbol{m}$ encodes the source radiation patterns as a double couple representing a point-wise kinematic discontinuity in the media.

The source position is represented by the coordinates $(x_s, y_s, z_s) \in \Omega$, not too close from the boundaries to avoid numerical issues due to absorbing boundary conditions. The position is chosen from a Latin Hypercube Sampling (LHS) with

$$x_s \in [1.2; 8.4\,km]$$
$$y_s \in [1.2; 8.4\,km]$$
$$z_s \in [-9.0; -0.6\,km] \tag{3}$$

In addition to the source position, the source is parametrized by the symmetric $3 \times 3$ moment tensor. An equivalent formulation is obtained with the three angles (strike, dip, rake) (Aki and Richards, 1980). With this representation, the angles are sampled from a LHS with a strike between $0°$ and $360°$, dip between $0°$ and $90°$, and rake between $0°$ and $360°$.

The source amplitude corresponds to a seismic moment $M_0 = 2.47 \times 10^{16}\,\mathrm{N\,m}$, and the source time evolution is a spice
bench given by $s(t) = 1 - \left(1 + \frac{t}{\tau}\right) e^{-\frac{t}{\tau}}$ with $\tau = 0.1\,\mathrm{s}$ (Fig. A1). Due to the linearity of the elastic wave equation (Eq. 1), it is important to notice that the choice of the source time function in the HEMEW$^S$-3D database does not constrain the variability of resulting ground motions.

First, any magnitude can be obtained by applying a scalar factor to the ground motion wavefields. Second, the response to any source time function can be computed from the Green function $\boldsymbol{G}(\boldsymbol{x}, t)$, which is the fundamental solution of the elastic
wave equation 1 when the source is an impulse point force located at $\boldsymbol{x}_s$ and occurring at $t = t_0$. For the reference source time function $s(t)$, the solution $\boldsymbol{u}(\boldsymbol{x}, t)$ provided in HEMEW$^S$-3D can be written as the convolution of the Green function with the source time function, $\boldsymbol{u}(\boldsymbol{x}, t) = \boldsymbol{G}(\boldsymbol{x}, t) * s(t)$

Computing the solution $u_1$ for a new source time function $s_1$ (provided that the moment tensor density $\boldsymbol{m}$ and the geological parameters $V_S$ remain the same) is straightforward following the steps below:

1. compute the Fourier transform of the reference source time function $\hat{s} := \mathcal{F}(s)$ and the solution $\hat{u} := \mathcal{F}(u)$

2. derive the Green function in the frequency domain $\hat{G} = \dfrac{\hat{u}}{\hat{s}}$

3. compute the Fourier transform of the new source time function $\hat{s}_1$

4. compute the new solution in the frequency domain $\hat{u}_1 = \hat{G} * \hat{s}_1$

5. deduce the new solution in the temporal domain $u_1 = \mathcal{F}^{-1}(\hat{u}_1)$

From these remarks, one should remember that ground motion wavefields in the HEMEW$^S$-3D database originate from point-wise sources with different locations $\boldsymbol{x}_s \in \mathbb{R}^3$ and orientations $\boldsymbol{\theta}_s \in \mathbb{R}^3$ but the same source time function.

## 3.3 Heterogeneous geological models

The HEMEW$^\text{S}$-3D database contains samples $\{V_S^{(i)}, \boldsymbol{x}_s^{(i)}, \boldsymbol{\theta}_s^{(i)}, \dot{\boldsymbol{u}}^{(i)}\}_i$ that satisfy equation 2 ($\dot{\boldsymbol{u}}$ denotes the velocity field obtained as the time derivative of the displacement field $\boldsymbol{u}$). The 3D geological models $V_S(\boldsymbol{x})$ are non-stationary random fields defined as a mean stair function (horizontal homogeneous layers) to which fluctuations are added, as illustrated in Figure 1.

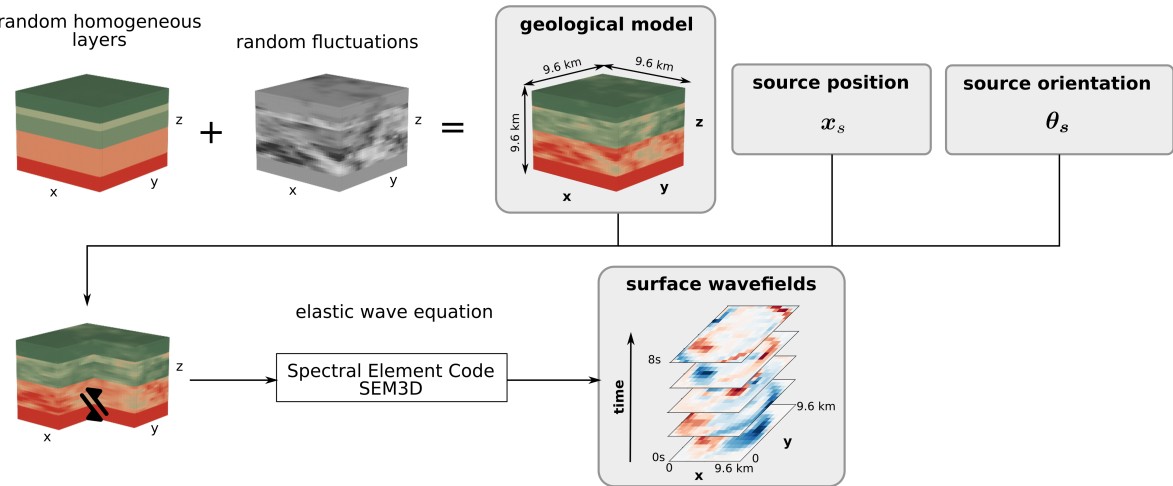

**Figure 1.** Geological models are built by adding heterogeneities to randomly chosen horizontal layers. Then, elastic waves are propagated from a source with a random position and random orientation to the surface, where velocity wavefields are synthetized.

### 3.3.1 Homogeneous models

A $1.8\,\text{km}$-thick homogeneous layer is imposed at the bottom of each geological model, with a $V_S$ value of $V_{S,max} = 4500\,\text{m/s}$. The minimum S-wave velocity is $V_{S,min} = 1071\,\text{m/s}$. Above the bottom layer, the number of horizontal layers and their thickness are randomly chosen for each sample $V_S^{(i)}$, with the sole constraint to fill the total depth with 2 to 7 layers. In particular, this means that velocity values are not sorted by depth. This choice is discussed in depth in Section 6. Then, the mean layer-wise value is drawn from the uniform distribution $\mathcal{U}([\mu_1; \mu_2])$. All layer-wise values are chosen independently. Values of $\mu_1 = V_{S,min}/0.6 = 1785\,\text{m/s}$ and $\mu_2 = V_{S,max}/1.4 = 3214\,\text{m/s}$ were determined to ensure that most values remain bounded within $[V_{S,min}; V_{S,max}]$ after the addition of random fields in each layer (see Table 2 for a summary of the parameters).

To recover the other geological properties, the ratio of P- to S-wave velocity was fixed to $V_P/V_S = 1.7$. The density $\rho$ is computed as a function of the P-wave velocity (Molinari and Morelli, 2011)

$$\rho = 1.6612 V_P - 0.4721 V_P^2 + 0.0671 V_P^3 - 0.0043 V_P^4 + 0.000106 V_P^5 \tag{4}$$

| Parameter | Statistical distribution |
|-----------|--------------------------|
| Number of heterogeneous layers $N_\ell$ | $\mathcal{U}(\{1,2,3,4,5,6\})$ |
| Layer thickness $h_1, \cdots, h_{N_\ell}$ | $\mathcal{U}(\{(h_1, \cdots, h_{N_\ell}) > 0 \mid h_1 + \cdots + h_{N_\ell} = 7.8\})$ |
| Mean $V_S$ value per layer | $\mathcal{U}([1785, 3214])$ |
| Layer-wise coefficient of variation | $\|\mathcal{N}(0.2, 0.1)\|$ |
| Layer-wise correlation length along x | $\mathcal{U}(\{1.5, 3, 4.5, 6 \text{ km}\})$ |
| Layer-wise correlation length along y | $\mathcal{U}(\{1.5, 3, 4.5, 6 \text{ km}\})$ |
| Layer-wise correlation length along z | $\mathcal{U}(\{1.5, 3, 4.5, 6 \text{ km}\})$ |

**Table 2.** Statistical distribution of each parameter describing the geological models. Mean $V_S$ values, coefficients of variation, and correlation lengths are chosen independently in each layer. Since the bottom layer has a constant thickness of $1.8\,\text{km}$, it is not included in these parameters.

Attenuation factors for P-waves ($Q_P$) and S-waves ($Q_S$) are computed as

$$Q_P = \max\left(\frac{V_P}{20}, \frac{V_S}{5}\right); Q_S = \frac{V_S}{10} \tag{5}$$

### 3.3.2 Addition of heterogeneities

The layers' thicknesses and mean values describe the general structure of the propagation domain and they correspond to the prior physical information usually available. However, geomaterials of the Earth's crust contain much variability, especially along the horizontal directions. This heterogeneity can be represented by random fields, characterized by their correlation length and coefficient of variation. Following previous studies on geological heterogeneity (e.g. Hartzell et al. (2010); Imperatori and Mai (2013); Khazaie et al. (2016); Scalise et al. (2021); Thompson et al. (2007)), we drew random fields with a von Karman correlation kernel and a Hurst exponent of $0.1$ (Chernov, 1960) (marginal distributions are log-normal to preserve positive values).

In order to provide a sufficient dataset variability, the choice of correlation lengths and coefficients of variation is tricky yet crucial (Colvez, 2021). The correlation length gives an idea of the distance above which two points $x_A$ and $x_B$ have independent geological properties $V_S(x_A)$ and $V_S(x_B)$. We chose correlation lengths randomly in $\{1.5, 3, 4.5, 6\}$ km, to mix samples with small- and large-scale heterogeneity. In addition, large coefficients of variation were chosen to provide high geological contrasts, following the folded normal distribution $\|\mathcal{N}(0.2, 0.1)\|$ with mean $0.2$ and coefficient of variation $0.1$. Coefficients of variation around $20\,\%$ are common at the surface (Arroucau, 2020), while it is known that values up to $40\,\%$ can be found locally (El Haber et al., 2021).

The 3D random fields computation is made highly efficient by the use of the spectral representation (Shinozuka and Deodatis, 1991; de Carvalho Paludo et al., 2019). With this formulation, a centered Gaussian random field $V_S$ determined by its auto-covariance function $\mathcal{R}$ can be decomposed as a sum of independent identically distributed random variables $(V_{S,n})_{-N \leq n \leq N}$,

with uniform distribution over $[0, 2\pi]$

$$V_S(x) = \sum_{n=-N}^{N} \sqrt{2\hat{\mathcal{R}}(n\Delta k)} \cos(n\Delta k \cdot x + V_{S,n})$$

where $\hat{\mathcal{R}}$ is the Fourier transform of the autocovariance function $\mathcal{R}$ and $\Delta k$ is the unit volume in $\mathbb{R}^3$.

Finally, $V_S$ values are clipped between $V_{S,min} = 1071\,\mathrm{m/s}$ and $V_{S,max} = 4500\,\mathrm{m/s}$. These bounds correspond to the velocity of shear-waves in hard sediments and at the bottom of the continental crust (Molinari and Morelli, 2011).

It should be noted that all layers have distinct coefficients of variation and correlation lengths, meaning that different random fields are drawn inside each layer. Also, random fields are drawn only once for each set of parameters.

### 3.3.3 Representation in the database

Geological realizations $V_S^{(i)}$ are discretized over a grid of $32 \times 32 \times 32$ elements (corresponding to $x$, $y$, $z$ axes) and are provided as `.npy` arrays. The total size of the geological dataset is 3.9 GB, split in 15 files of 2000 geological models for easier data management. Additionally, metadata give parameters of each layer: the mean $V_S$ value, the thickness, the coefficient of variation, and the correlation lengths along $x$, $y$, and $z$.

### 3.4 Solutions of the wave equation

The elastic wave equation was solved in each domain $i$ by means of the open source code SEM3D[1] (Touhami et al., 2022) based on the Spectral Element Method (Faccioli et al., 1997; Komatitsch and Tromp, 1999). The dimension of the simulation mesh is prescribed by the maximum frequency $f_{max}$ one aims at exactly resolving. In this study, $f_{max}$ was fixed at 5 Hz, which is relatively high for this type of simulations. Many simulations have been conducted so far with an accuracy up to 1 or 2 Hz
(Rekoske et al., 2023; Rosti et al., 2023), while high-fidelity simulations for local realistic earthquake scenarios extend up to 10 Hz (Castro-Cruz et al., 2021; De Martin et al., 2021; Heinecke et al., 2014) (and exceptionally up to 18 Hz, such as in Fu et al. (2017)). Then, the smallest wavelength $\lambda_{min} = V_{S,min}/f_{max}$ must be described on the mesh by at least 5 quadrature points. With 7 Gauss-Lobatto-Legendre quadrature points per mesh element, this leads to elements of size $h = \frac{7}{5} \cdot \frac{V_{S,min}}{f_{max}} = 300\,\mathrm{m}$. This explains that 32 elements in each direction amount to a domain size of $L = 9.6\,\mathrm{km}$. The time-marching scheme is a leap-frog
second-order accurate explicit scheme, solved for velocity fields.

To maintain reasonable computational loads and reflect realistic scenarios, velocity fields were recorded only at the surface of the propagation domain. A regular grid of $32 \times 32$ sensors was placed between $150\,\mathrm{m}$ and $9450\,\mathrm{m}$ in both horizontal directions (implying a distance of $300\,\mathrm{m}$ between two neighbouring sensors). At each monitoring point, the three-component velocity field is synthetized with a $100\,\mathrm{Hz}$ sampling frequency between $0\,\mathrm{s}$ and $8\,\mathrm{s}$. Although the sampling frequency is higher than
210 the Nyquist frequency (i.e. $2 \times f_{max} = 10\,\mathrm{Hz}$), the value of $100\,\mathrm{Hz}$ was chosen to match the temporal resolution of recorded time series in several publicly accessible datasets (e.g. STEAD (Mousavi et al., 2019), INSTANCE (Michelini et al., 2021)). The sampling frequency is sufficient to allow an accurate computation of Peak Ground Velocity (PGV), derive the acceleration

---

[1]https://github.com/sem3d/SEM

time series with finite differences, and compute the Peak Ground Acceleration (PGA). Figure 2 illustrates velocity waveforms at eight virtual sensors.

### 3.4.1 Representation in the database

Velocity fields are provided as `.h5` files. Each file contains three keys: `uE, uN, uZ` corresponding to the three components of ground motion (East-West, North-South, Vertical). Each velocity field is of shape $32 \times 32 \times 800$ where the first index corresponds to the $y$ axis, the second index to the $x$ axis, and the third index to the temporal axis.

Files are gathered in `.zip` archives containing 100 simulation results. The 300 .zip files amount to 263.4GB. They are downloadable individually (0.87GB per file).

## 4 Dataset analysis

### 4.1 Descriptive statistics of the temporal evolution

Since most of the geological parameters are chosen uniformly randomly (Table 2), the geological dataset is well-balanced: geological models with 1 to 6 layers are equipartitioned and all random fields parameters have approximately the same frequency. Mean $V_S$ values range from $1756 \, \text{m/s}$ to $3145 \, \text{m/s}$.

The first wave arrival time is a crucial parameter for earthquake early warning and seismic phase picking is a common task with deep learning models. Arrival time depends on the distance between the earthquake source and the monitoring sensor, as well as the geological properties on the propagation path. Wave arrival times are usually determined from recordings, either manually by experts, or with machine learning methods. However, it is possible to compute almost exact arrival times from synthetic velocity fields since ground motion is almost zero before the first wave arrival. Therefore, we obtained wave arrival times for P-waves as the earliest time-step where the amplitude exceeds $0.1 \, \%$ of the maximum amplitude. Due to the source depth variability and the different wave velocities in the geological models, first wave arrival times vary significantly among samples and among sensors. Figure 3a shows that $10 \, \%$ of velocity time series are initiated before $0.65 \, \text{s}$ while $10 \, \%$ of time series are still null after $2.17 \, \text{s}$.

As expected, the P-wave arrival time is strongly correlated with the hypocentral distance (Fig. 4a) since shorter hypocentral distances are associated with shorter propagation paths. The mean velocity on the propagation path also influences the first wave arrival time but variability is higher. Figure 4b indeed shows that the P-wave arrival time is negatively correlated with the mean S-wave velocity in the whole domain. It confirms that waves propagate slower when the mean velocity is lower. The mean velocity gives an approximation of the velocity values encountered by the waves along the propagation path. In particular, the mean velocity does not depend on the sensor position with this approximation.

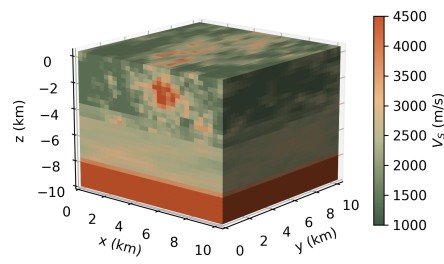

## Velocity fields at x=8.55km

| East-West | North-South | Vertical |
| --- | --- | --- |

**Figure 2.** 3-component velocity waveforms synthetized at eight virtual sensors on a line parallel to the $y$ axis at $x$=8.85 km. The shaded area extends from 5 % to 95 % of the Arias intensity, hence its length equals the Relative Significant Duration (RSD). The corresponding geological model is shown at the top. The source is located at $(2.04\,\mathrm{km}, 3.64\,\mathrm{km}, -2.17\,\mathrm{km})$.

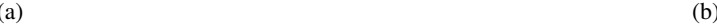

 

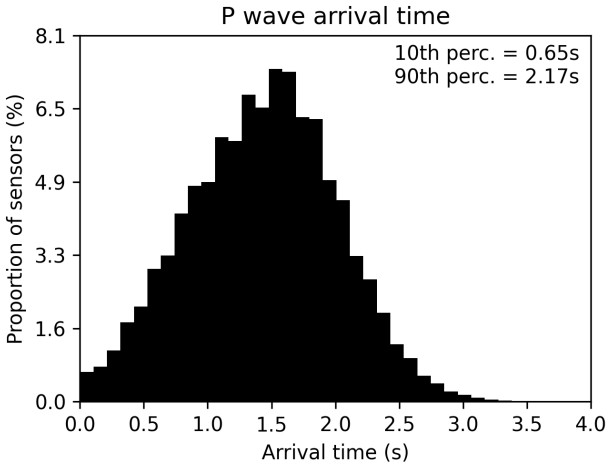
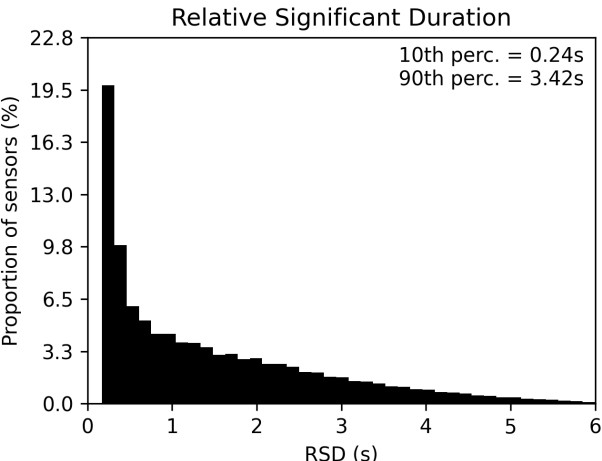

**Figure 3.** Distributions of the temporal features of velocity time series at each monitoring sensor and for $30\,000$ samples. (a) the first P-wave arrival time is computed on the vertical component (b) the Relative Significant Duration (RSD) is shown for the East-West component, results are very similar for the two other components

The temporal evolution of ground motion can also be characterized by its Relative Significant Duration (RSD). It corresponds to the duration of the signal between 5% and 95% of the Arias intensity ($I_A$) (Arias, 1970)

$$I_A = \frac{\pi}{2g} \int_0^T a^2(t)dt \tag{6}$$

where $a(t)$ is the acceleration field and $T$ is the total duration of the signal. Figure 3b shows that RSD covers a large variation
range, from $0.17\,\text{s}$ to $7.60\,\text{s}$. This variability is illustrated in Fig. 2 where the grey areas represent the RSD. One can especially notice that samples with a strong pulse have a small RSD. Indeed, most of the energy is concentrated around the pulse.

Quantitatively, the HEMEW$^\text{S}$-3D database contains short ground motions since $10\,\%$ of time series have a RSD lower than $0.24\,\text{s}$, as well as longer ground motions where $10\,\%$ of time series have a RSD longer than $3.42\,\text{s}$. The median RSD is $1.06\,\text{s}$. These short RSD values are related to the absence of high-frequency components in the coda and the dominance of high pulse-
250 like time series in cases with shallow sources and low heterogeneity contrasts. Combined with the short P-wave arrival times, RSD values also justify that the $8\,\text{s}$ window contains the significant part of ground motion.

### 4.2 Descriptive statistics related to energy content

The PGV is computed as the maximum absolute value over all timesteps separately on each component. The PGV is slightly lower on the vertical component than the two horizontal components (Fig. 5). It is very similar between the East-West and

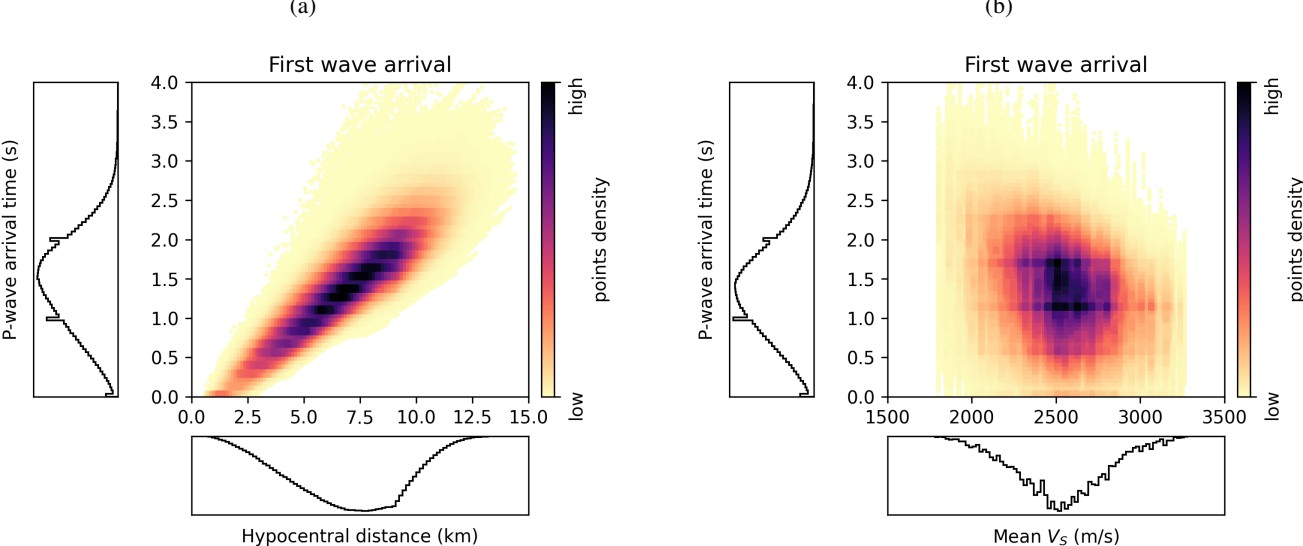

(a)                                                     (b)

**Figure 4.** For each sample and each sensor, the P-wave arrival time is shown against (a) the hypocentral distance, (b) the mean S-wave velocity in the 3D domain.

North-South components, which is expected since the HEMEW$^S$-3D database is statistically invariant per horizontal rotation. Figure 5 shows that the PGV extends over three orders of magnitude, with the first percentile being equal to $0.89\,\mathrm{cm/s}$ while the 99th percentile equals $129.3\,\mathrm{cm/s}$. The median PGV is $8.9\,\mathrm{cm/s}$. The 1st- to 99th-percentile interval is in line with ground motion observed within a few kilometers of moderate-magnitude earthquakes (e.g. Convertito et al. (2022)).

When the propagation path is longer, seismic waves encounter more geological heterogeneities. They create a dispersion
and diffraction of waves that spread the energy signal over time. Larger hypocentral distances are associated with longer propagation paths. Figure 6a then shows that the PGV is negatively correlated with the hypocentral distance.

It is also known that the seismic energy $E_s$ generated by a fault rupture is

$$E_s = \frac{M_0 \Delta\sigma}{2\mu} \tag{7}$$

where $M_0$ is the seismic moment, $\Delta\sigma$ is the stress drop and $\mu$ is the shear modulus at the fault location. Knowing that the shear
wave velocity writes $V_S = \sqrt{\mu/\rho}$, equation 7 indicates that the seismic energy is inversely proportional to $V_S^2$. And Figure 6b confirms that the PGV is negatively correlated with the velocity of S-waves at the source location.

### 4.3   Distribution of Pseudo-Spectral Acceleration (PSA)

The Pseudo-Spectral Acceleration (PSA) is a commonly used metric to estimate structural response. It evaluates the maximal acceleration of a one-degree-of-freedom oscillator (with a $5\,\%$ damping) with a natural period $T$. At $T$=$0.2\,\mathrm{s}$, the PSA in
the HEMEW$^S$-3D database is comprised between $2.3 \times 10^{-3}\,\mathrm{g}$ and $81.2\,\mathrm{g}$. It decreases to the interval $5.8 \times 10^{-4}\,\mathrm{g}$ ; $6.1\,\mathrm{g}$ at

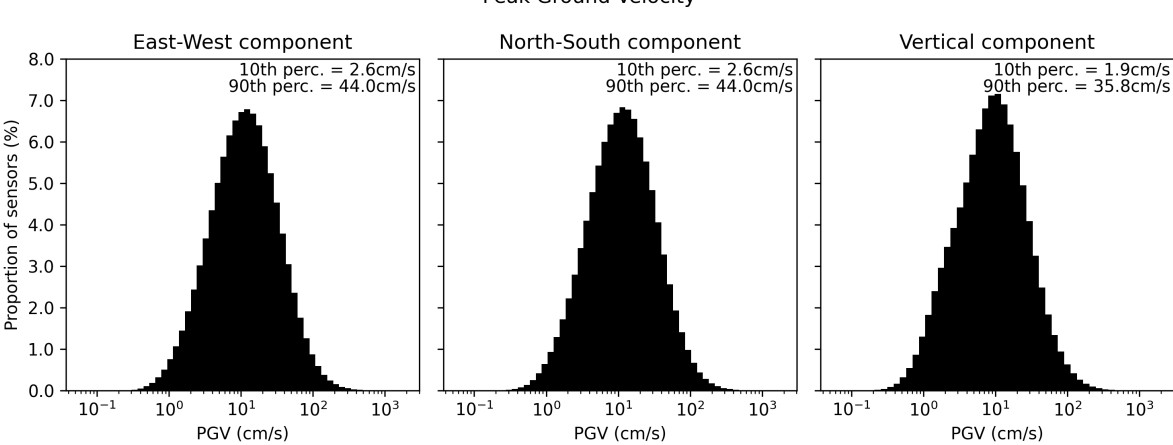

**Figure 5.** The Peak Ground Velocity (PGV) is computed as the the maximum absolute value over all timesteps separately on each component. There is one value for each of the $32 \times 32$ sensors and each of the $30\,000$ samples.

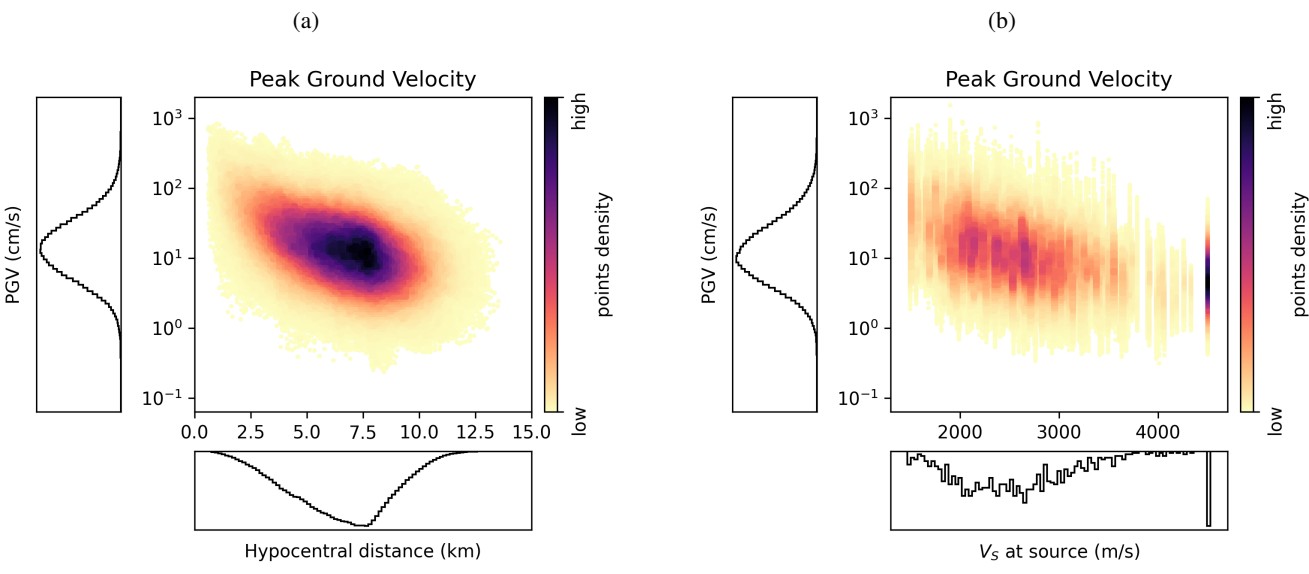

**Figure 6.** For each sample and each sensor, the PGV is shown against (a) the hypocentral distance, (b) the S-wave velocity at the source location. The PGV is computed on the East-West component, results are very similar for the two other components.

$T$=1.0 s. Figure 7 additionally shows that there exists a negative correlation between the PSA and the hypocentral distance. The distance-dependent PSA values can be compared with existing Ground Motion Models (GMMs), as shown in Fig. 8.

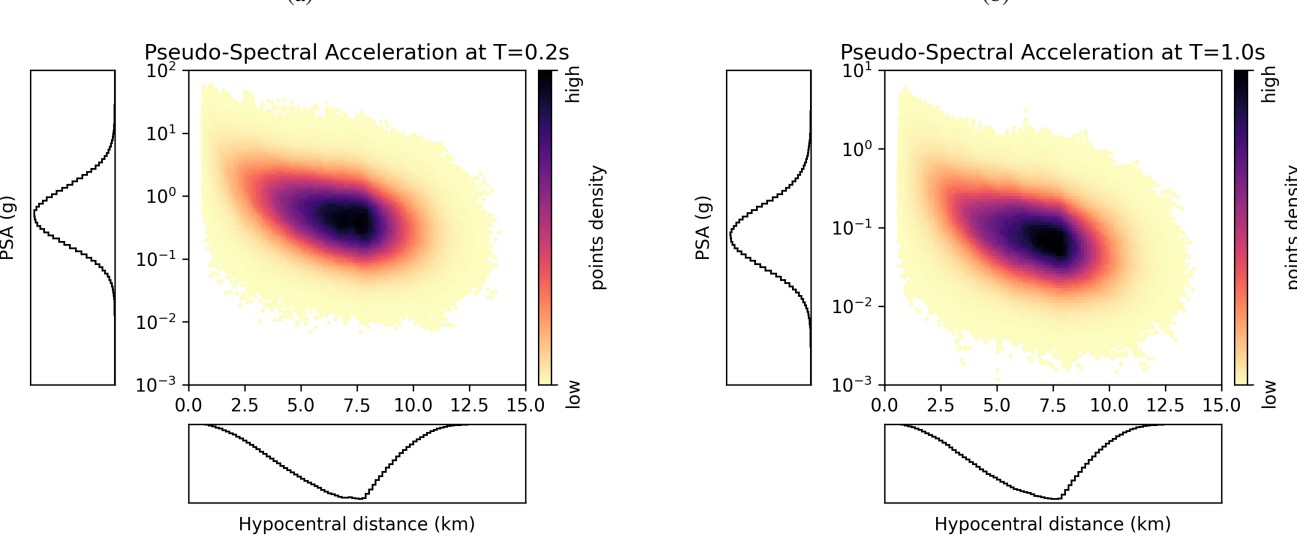

**Figure 7.** For each sample and each sensor, the Pseudo-Spectral Acceleration is shown against the hypocentral distance at period $T$=0.2 s (a) and $T$=1.0 s (b). The PSA is computed on the East-West component, results are very similar for the two other components.

GMMs provide analytical formula to compute intensity measures, like PSA and PGV, based on regression analyses. They are mainly derived from databases of recorded earthquakes, although numerical simulations can also be used. The PSA estimated from the HEMEW$^\text{S}$-3D database is compared with four GMMs (all taken with a moment magnitude $4.9$ that corresponds to the seismic moment of the HEMEW$^\text{S}$-3D database):

1. GMM from Atkinson and Boore (2006), computed with $V_{S,30}$=2000 m/s

2. GMM from Atkinson (2015), computed with a depth of $1$ km (solid line in Fig. 8) and a depth of $7.5$ km (dashed line in Fig. 8)

3. GMM from Chiou and Youngs (2014), computed with $V_{S,30}$=2000 m/s, a depth of $1$ km, and different dip and rake angles (those two last factors having little influence on the PSA)

4. GMM from Shahjouei and Pezeshk (2016)

Figure 8 shows that the mean PSA computed from the HEMEW$^\text{S}$-3D database is in good agreement with all GMMs and the standard deviation is smaller than GMMs. The Atkinson (2015) GMM illustrates the influence of depth, especially for the smallest hypocentral distances. The HEMEW$^\text{S}$-3D PSA values fit better with the GMM PSA for shallow sources (solid purple line, 8) than deeper sources (dashed purple line).

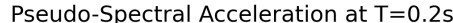

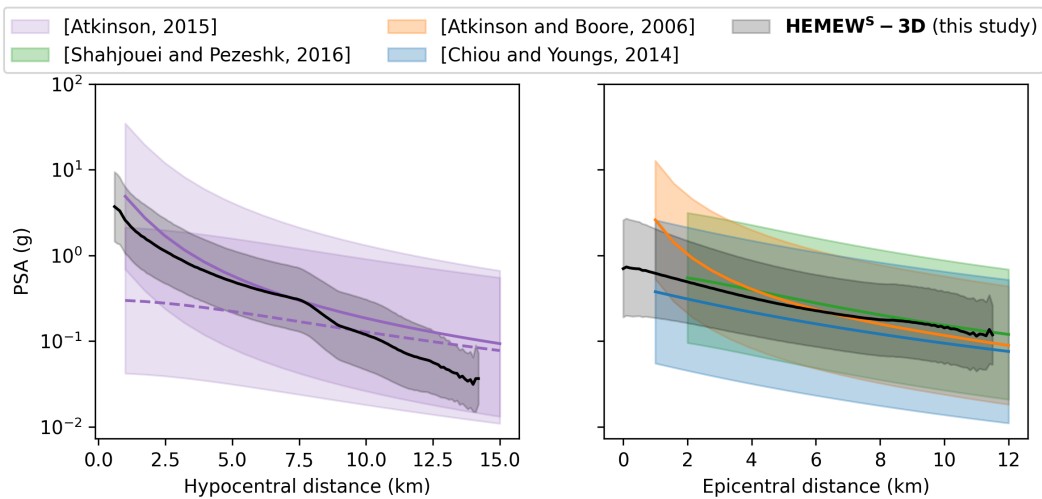

**Figure 8.** Horizontal PSA at period T=0.2 s as a function of hypocentral distance (left) and epicentral distance (right) for GMMs by Atkinson (2015) with a depth of 1 km (purple solid line, left), Atkinson (2015) with a depth of 7.5 km (purple dashed line, left), Atkinson and Boore (2006) (orange, right), Chiou and Youngs (2014) (blue, right), Shahjouei and Pezeshk (2016) (green, right), and our HEMEW[S]-3D database (black). Solid lines correspond to the mean PSA and shaded areas to one standard deviation. Horizontal PSA is computed as the geometrical mean of East-West and North-South components.

## 4.4 Dimensionality

In supervised deep learning, it is always challenging to determine whether the size of the database (i.e. the number of samples) is sufficient to represent its variability. This questions relates to the definition of the intrinsic dimension of the dataset, which indicates the number of hidden variables that should be necessary to represent the main features of the samples. In the following, we provide insights on this question with the intrinsic dimension based on the Principal Component Analysis (Section 4.4.1), the correlation dimension (Section 4.4.2), the Maximum Likelihood Estimate (Section 4.4.3), and the Structural Similarity Index (Section 4.5).

### 4.4.1 Principal Component Analysis (PCA)

The Principal Component Analysis (PCA) decomposes data in principal components that correspond to the directions where data vary the most. For different sizes of datasets, we compute the number of principal components required to retain $95\,\%$ of variance and define this number as the intrinsic dimension of data. The 3D geological models and the 3D ground motion wavefields are transformed into 1D vectors to perform the PCA. To reduce the memory requirements, ground motions are analyzed only on the East-West component. Geological models are represented by $32 \times 32 \times 32 = 32\,768$ points and ground

motions contain $16 \times 16 \times 320 = 81\,920$ points (16 sensors in directions $x$ and $y$, and 320 time steps between $0\,\text{s}$ and $6.4\,\text{s}$). To ease the computation on the large sample covariance matrix, an incremental PCA algorithm was used (Ross et al., 2008).

Table 3 and Figure B1 show that more than 1000 principal components are needed to reconstruct the geological models with high accuracy whereas the intrinsic dimension of ground motion wavefields is around 4900. It is reasonable that the wavefields intrinsic dimension is larger than the geological dimension since wavefields variability is created by geological variations and the source position. However, due to its linearity, the PCA requires a large number of components to accurately represent complex patterns. Therefore, it may overestimate the intrinsic data dimension.

**Table 3.** Database intrinsic dimension estimated by PCA, correlation dimension, and MLE for the geological database and the velocity fields database, depending on the number of data samples

| | Geological database | | | | | Velocity fields database | | | | |
|---|---|---|---|---|---|---|---|---|---|---|
| Nb. of samples ($\times 10^3$) | 2 | 6 | 10 | 20 | 30 | 2 | 6 | 10 | 20 | 30 |
| PCA | 491 | 766 | 880 | 1006 | 1094 | 853 | 2057 | 2853 | 4073 | 4875 |
| Correlation dimension | 8.8 | 8.3 | 8.4 | 8.3 | 8.2 | 2.2 | 2.3 | 2.3 | 2.3 | 2.3 |
| MLE | 17.9 | 23.4 | 26.7 | 31.5 | 33.9 | 107.2 | 129.0 | 116.6 | 113.8 | 110.9 |

#### 4.4.2 Correlation dimension

An alternative dimensionality measure was introduced by Grassberger and Procaccia (1983) as the correlation dimension, which characterizes the distance between pairs of samples. For a dataset of $N$ samples $\{V_S^{(i)}\}_{1 \leq i \leq N}$ and a given *radius* $r$, the correlation dimension ($C_N(r)$) is defined as the ratio of sample pairs $(V_S^{(i)}, V_S^{(j)})_{i \neq j}$ being at distance less than $r$

$$C_N(r) = \frac{2}{N(N-1)} \sum_{i=1}^{N} \sum_{j=i+1}^{N} \mathbb{1}\left( \|V_S^{(i)} - V_S^{(j)}\| \leq r \right) \tag{8}$$

Figure B2 indicates a correlation dimension of 8 for the geological dataset, which is significantly lower than the PCA dimension. In fact, it is known that the correlation dimension may underestimate the intrinsic dimension, especially "when data are scattered" (Qiu et al., 2023), which is likely to be the case in high-dimensional spaces. However, the correlation dimension of the ground motion wavefields is debatable as it drops to 2. Figure B3c shows that the log-log representation of $C_N(r)$ does not produce an obvious linear part, which makes it difficult to identify the correlation dimension.

#### 4.4.3 MLE intrinsic dimension

Levina and Bickel (2004) proposed another approach based on the Maximum Likelihood Estimator (MLE) of the distance to the closest neighbours. Figure B4 shows the evolution of the intrinsic dimension as a function of the number of samples for geological models and velocity wavefields. The intrinsic dimension of geological models is 34 while the dimension of velocity

wavefields is larger (around 110). Although this method may still underestimate data with high intrinsic dimensionality (Qiu et al., 2023), it provides higher estimates than the correlation dimension.

It can also be noted that the intrinsic dimension increases with the number of samples, as was observed for the PCA. This may reflect a flaw in the intrinsic dimension's definition or it may indicate that despite being already large, our database of 30 000 samples does not capture all the variability.

## 4.5 Structural similarity

The correlation dimension is computed from the Euclidean distance between pairs of geological models. However, point-wise metrics do not necessarily best represent similarities between geological models, and alternative metrics such as the Structural Similarity Index Measure (SSIM) have been introduced for this purpose (Wang et al., 2004). This index theoretically ranges from 0 to 1, with 0 indicating no similarity and 1 indicating perfectly similar geological models (although values between -1 and 0 can be obtained numerically from the covariance computation). The SSIM of two geological models $A$ and $B$ is defined as

$$SSIM(A, B) = \frac{(2\mu_A\mu_B + C_1)(2\sigma_{AB} + C_2)}{(\mu_A^2 + \mu_B^2 + C_1)(\sigma_A^2 + \sigma_B^2 + C_2)} \tag{9}$$

where $\mu_A$ and $\mu_B$ are the means of $A$ and $B$, $\sigma_A$ and $\sigma_B$ are the unbiased estimators of the variance of $A$ and $B$, $\sigma_{AB}$ is the unbiased estimator of the covariance of $A$ and $B$, $C_1$ and $C_2$ are constants determined from the range of $A$ and $B$ values.

Figures B5a and B5b illustrates two pairs of geological models with the same SSIM of $0.6$, meaning rather high similarity. The first geologies have similar mean values but different heterogeneities, resulting in a low Euclidean distance (Fig. B5a) while the second geologies have different mean values, leading to a higher Euclidean distance (B5b).

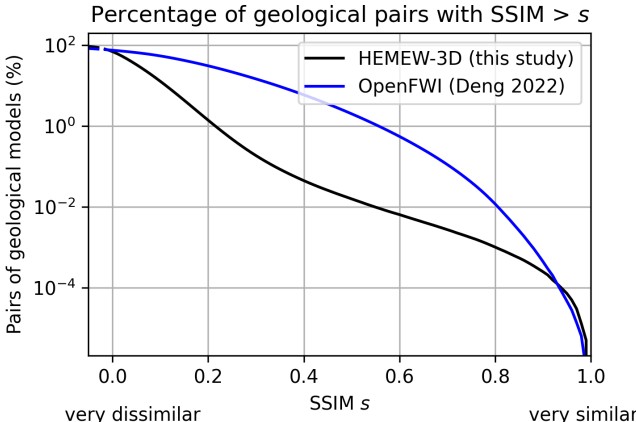

**Figure 9.** The Structural Similarity Index Measure (SSIM) quantifies the visual ressemblance between images, in a way that should mimic human perception. For each SSIM value $s$ on the $x$-axis, the percentage of geological pairs being more similar than $s$ is reported on the $y$-axis.

To give insights on the sparsity of the geological database, Figure 9 shows that only $1.4\%$ of geological pairs have a SSIM greater than $0.2$. This means that geological models are generally very distinct from each other in the HEMEW$^S$-3D database. For comparison, the 2D OpenFWI dataset leads to significantly higher SSIM, with $31\%$ of geologies having a SSIM larger than $0.2$ (3000 models were chosen from each of the 10 OpenFWI families, (Deng et al., 2022)).

## 5 Applications

### 5.1 Dimensionality reduction of geological models

Dimensionality analyses have shown that at least $1000$ principal components are necessary to represent geological models with enough accuracy, as measured by the reconstructed variance. This means that the PCA provides a basis of 3D models to decompose a wide diversity of geological models. One can consider geological models that are very different from the random fields contained in the HEMEW$^S$-3D database, for instance embedding a basin shape.

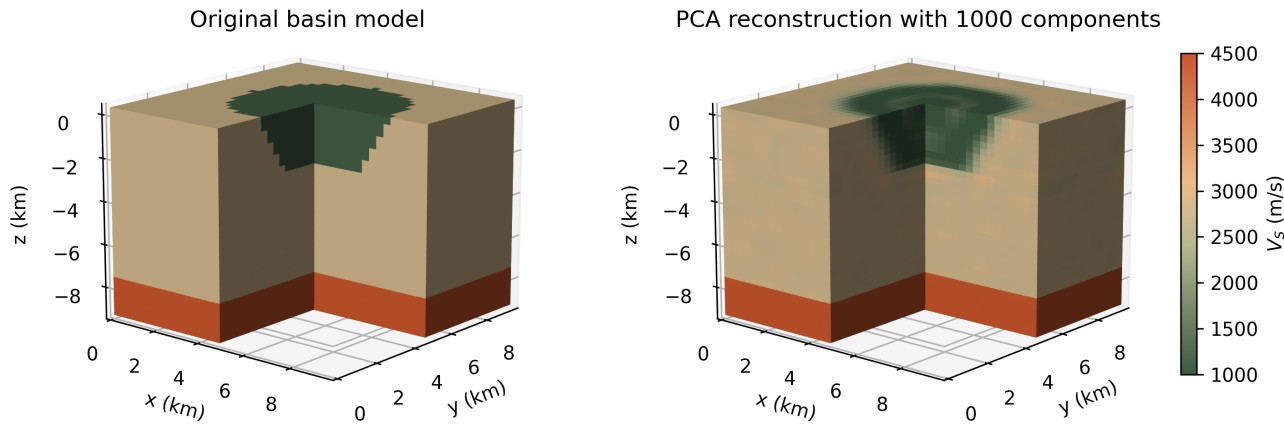

**Figure 10.** The original geological model (left) contains two homogeneous layers and a circular basin inserted inside the top layer. The PCA reconstruction was obtained with 1000 PCA components (right).

Figure 10 shows that 1000 principal components allow a good reconstruction of the basin shape with correct velocity values inside and outside the basin. Edges are slightly blurred, which is expected since sharp contrasts correspond to high spatial frequencies that require many principal components. This example illustrates the generalization ability of the HEMEW$^S$-3D database from a geometrical point of view. To match the design of the HEMEW$^S$-3D database, the velocity values are chosen within the same bounds. If one were to consider real sedimentary basins, rescaling should be applied to target lower velocity values.

The influence of the PCA reconstruction on the generated velocity wavefields was investigated in more details in Lehmann et al. (2022). It was shown that wavefields created by the propagation of seismic waves inside the reconstructed geological

models and the reference model are very similar. When the initial geological model has strong heterogeneities, heterogeneities tend to be blurred in the PCA reconstruction, which reduces the dispersion of seismic waves. As a consequence, velocity wavefields generated inside the reconstructed geological model have slightly larger amplitudes.

## 5.2 Velocity fields predictions

Since the HEMEW[S]-3D database associates geological models and sources with their corresponding velocity wavefields, it can serve to predict the latter from the former. Neural operators are one class of SciML models that have shown great success in the prediction of parametric PDEs. One can mention in particular the Multiple Input Fourier Neural Operator (MIFNO, Lehmann et al. (2024)) that uses the Fast Fourier Transform to learn the frequential representation of the elastic wave equation and a dedicated handling of the source term (Fig. C1).

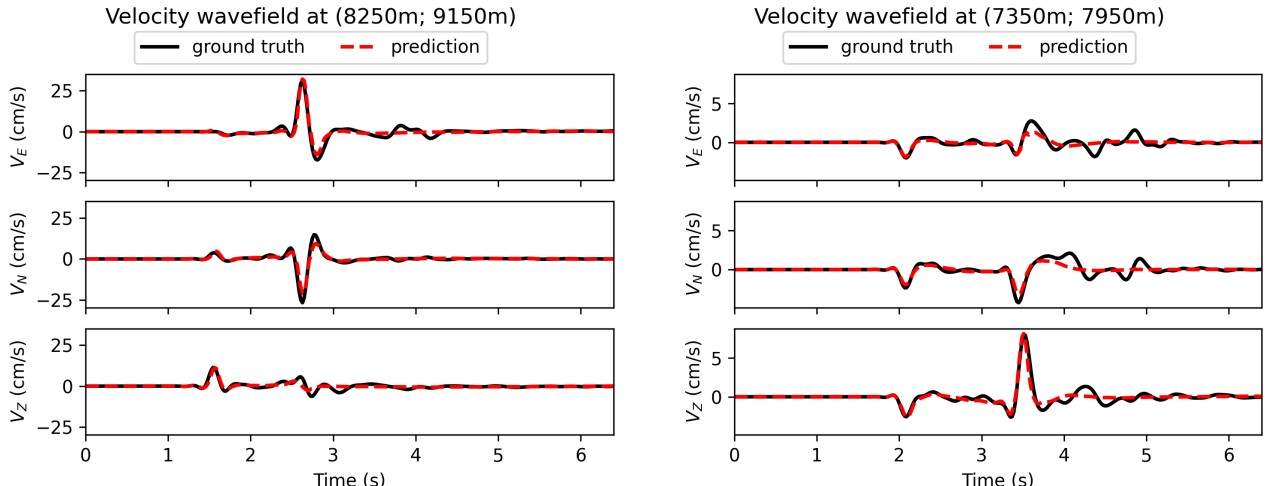

**Figure 11.** For two spatial points, the velocity field predicted by the MIFNO (red dashed line) is compared with the reference from the HEMEW[S]-3D database (black line).

For each geological model and source in the HEMEW[S]-3D database, the MIFNO predicts the velocity field at each surface point. Figure 11 illustrates that the MIFNO gives accurate predictions for samples with different ground motions. This shows that the variability and size of the HEMEW[S]-3D database are appropriate to train complex SciML models.

## 5.3 Other potential applications

Thanks to the large number of simulations, one can also envision studying the variability of ground motion to capture its statistical distribution. In particular, one can investigate the best sampling that minimizes the number of samples while preserving the largest ground motion variability (Tarbali and Bradley, 2015).

The surface ground motion can also be considered as an "outcropping bedrock" response which is classically used in 1D site-effect and soil-structure interaction analyses, and which may require deconvolution.

## 6 Limitations and perspectives

Since HEMEW$^S$-3D is the first database providing 3D ground motion, it is constrained by some hypotheses to control its size and allow machine learning applications.

First, the minimum S-wave velocity of $1071\,\mathrm{m/s}$ is rather high when compared to S-wave velocities in soft sediments (typically of few hundreds of m/s) but coherent for hard sediments (Molinari and Morelli, 2011). $V_S$ values in the HEMEW$^S$-3D database are also in line with national velocity models with a low spatial resolution that display surface values around $2000\,\mathrm{m/s}$. The choice of minimum $V_S$ implies in particular that the HEMEW$^S$-3D dataset is not targeted towards site-effects analyses in sedimentary basins. One should also note that the vertical resolution of the geological models is $300\,\mathrm{m}$ while very low $V_S$ values are more commonly encountered in the first tens of meters. These low values would be averaged with higher deeper values in our models. In particular, this means that the upper velocity in the HEMEW$^S$-3D database should be understood as the average velocity between $0\,\mathrm{m}$ and $-300\,\mathrm{m}$. It is not comparable with the common notion of $V_{S,30}$ (average $V_S$ in the first $30\,\mathrm{m}$). Reducing the minimum velocity poses no theoretical limitation but would increase the computational cost of the subsequent numerical simulations since it increases the number of mesh elements.

Second, the maximum S-wave velocity of $4500\,\mathrm{m/s}$ corresponds to existing $V_S$ values at the bottom of the Earth's crust often adopted in velocity models (Duverger et al., 2021; Molinari and Morelli, 2011). In addition, the bottom layer has a fixed thickness and value that originates from earlier works. Therefore, variability is considered only above this constant layer.

Third, we do not constrain the ordering of layer-wise $V_S$ values to provide a large database variability that is essential for machine learning perspectives. Similar choices were done in the widely-used OpenFWI database (Deng et al., 2022), in which the Flat-Vel-B family is based on a random arrangement of layers. This means that some layer arrangements may be unphysical, for instance if the mean values are linearly decreasing with depth. However, it is important to notice that the physics of wave propagation is still satisfied in those situations, which is the main concern of this work. One choice opposite to ours would be to impose layer-wise values increasing with depth (as done in the OpenFWI Flat-Vel-A family for instance). However, this would remove all models with velocity inversion, which can be found in complex geological contexts. From the metadata provided, users can filter geological models with custom criteria to exclude those *outliers* from their study. For instance, one custom criterion could be the impedance contrast, defined as the ratio between impedance ($Z = V_S \times \rho$) in one geological layer $Z_\ell$ and the layer above $Z_{\ell-1}$. Figure A2 illustrates the distribution of impedance contrasts and shows that they are realistic. 9028 geological models have a minimum impedance contrast smaller than $0.7$.

Additionally, more diverse configurations could be designed by relaxing the assumption that all geological parameters depend on a single variable. This would imply, for instance, varying the $V_P/V_S$ ratio. Random anisotropic heterogeneities can also be generated for more diversity (Ta et al., 2010).

The domain size was limited to $9.6\,\mathrm{km}$ to prove that SciML was possible with a manageable dataset size. This size allows reasonable local studies and is already larger than existing 2D databases (Tab. 1). Extending the spatial size is certainly of interest for some seismological applications and requires additional computational costs. In summary, with larger computational budgets/lesser memory constraints, it would become possible to

- consider larger and deeper geological models,

– design models with a higher spatial resolution,

- include lower minimum $V_S$ ,

- increase the frequency limit of the wave propagation simulations,

- increase the spatial sampling of virtual sensors,

- increase the temporal duration of signals, to match the longer epicentral distances coming from larger models.

It should also be noted that numerical simulations are only valid up to a $5\,\mathrm{Hz}$ frequency, due to the mesh design, with numerical pollution for frequencies larger than $5\,\mathrm{Hz}$. We observed that it is crucial to apply a low-pass filter (with a cutoff frequency of $5\,\mathrm{Hz}$) to the velocity fields before using machine learning models, otherwise the model may try to fit numerical noise.

## 7 Conclusions

We presented the HEMEW$^S$-3D database (HEterogeneous Materials and Elastic Waves with Source variability) that contains $30\,000$ geological models, source parameters and the time- and space-dependent surface wavefields generated by the propagation of seismic waves through each geological model. This database was conceived for the forward problem of wave propagation.

  Geological models are built from horizontal layers randomly arranged and they correspond to the velocity of shear waves
($V_S$). They represent a domain of size $9.6\,\mathrm{km} \times 9.6\,\mathrm{km} \times 9.6\,\mathrm{km}$ discretized in $300\,\mathrm{m}$-wide elements. $V_S$ values are comprised between $1071\,\mathrm{m/s}$ and $4500\,\mathrm{m/s}$. Then, random fields are added independently in each geological layer to create 3D heterogeneities. Their parameters (coefficients of variation and correlation lengths) vary widely to cover diverse geological configurations and are given as metadata. Geological models are provided as cubes with $32 \times 32 \times 32$ voxels.

  Seismic waves propagate numerically from the earthquake source to the surface. Point-wise sources have a random position
and orientation. They are synthetized at the surface of the propagation domain by a grid of $32 \times 32$ sensors for $8\,\mathrm{s}$. Simulations are conducted with the High-Performance Computing code SEM3D and amount to a total computational time of $9 \times 10^5$ core.hours. The dataset description shows that the $8\,\mathrm{s}$ time window covers most significant ground motion at the surface.

  Ground motion characteristics differ strongly between samples. They were analyzed in terms of Relative Significant Duration (RSD), P-wave arrival time, Peak Ground Velocity (PGV), and Pseudo-Spectral Acceleration (PSA). In addition to

quantifying the distributions of essential intensity measures in seismology, these analyses confirm expected relationships between physical parameters and ground motion characteristics. In particular, hypocentral distance, $V_S$ at the source location, and mean velocity were investigated. Comparisons with Ground Motion Models (GMMs) show that PSA is comparable with estimates from recorded earthquakes. This indicates the usefulness of the HEMEW$^S$-3D database to complement databases of recorded earthquakes.

Due to the size of individual samples in the HEMEW$^S$-3D database, one may wonder whether data could be represented with less parameters to reduce memory requirements. To this end, we explored different methods to estimate the data intrinsic dimension and we exemplified the well-known fact that they can lead to very different values. Taking the MLE as a lower bound, one can argue that the intrinsic dimension of the geological database is at least 30. In addition, the low values of the SSIM indicate that geologies are sparse and quite distant from each other in the HEMEW$^S$ database.

Concerning the velocity wavefields, the PCA and the MLE confirm the intuition that the intrinsic dimension is larger than the geological dimension since the source adds variability to the time arrival of wavefields as well as their location at the surface. In this situation, it is reasonable to consider that the intrinsic dimension of ground motion is at least on the order of 100. However, if data are decomposed with the PCA, then the number of principal components is a few thousands. The correlation dimension yields questionable estimates of the intrinsic dimension that contradict our intuition and the PCA and MLE outcomes.

By providing a large number of physics-based simulations, the HEMEW$^S$-3D database offers new perspectives to study the relationship between geological properties and surface ground motion. It led to the first neural operator predicting 3D ground motion but many applications, in statistics, scientific machine learning, and deep learning are envisioned. We designed the database to be as generic as possible and we believe that several scientific communities can benefit from it.

## 8   Code and data availability

The database is referred to as Lehmann (2023) and can be downloaded at https://doi.org/10.57745/LAI6YU. The wave propagation code SEM3D is available at https://github.com/sem3d/SEM. The code used to generate the HEMEW$^S$-3D database is given at https://github.com/lehmannfa/HEMEW3D.

## Appendix A: Dataset description

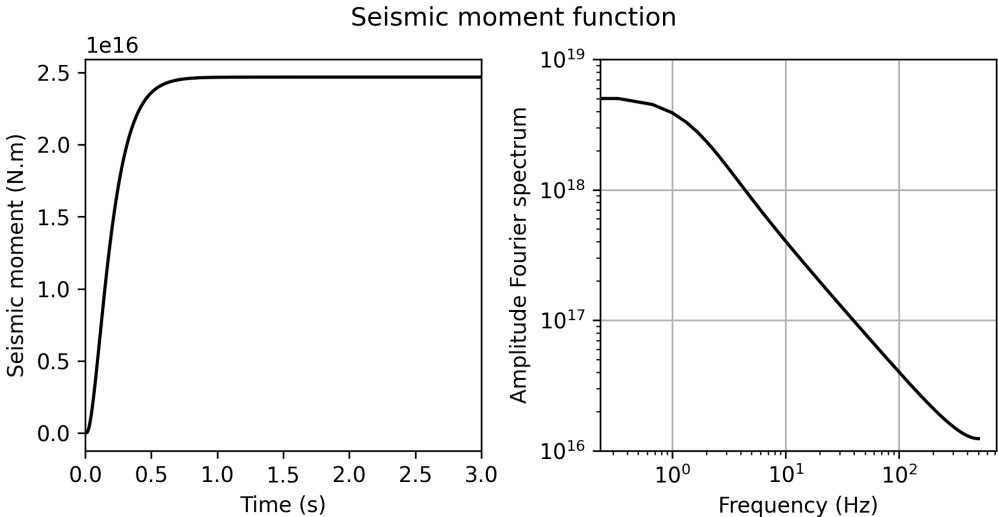

**Figure A1.** The seismic moment function in the HEMEW$^S$-3D database

## Distribution of impedance constrasts
## between two consecutive layers

**Figure A2.** Distribution of impedance contrasts in the HEMEW$^S$-3D database. The impedance contrast is computed as the ratio between impedance in one layer ($Z_\ell = V s_\ell \times \rho_\ell$) and impedance in the layer above ($Z_{\ell-1}$).

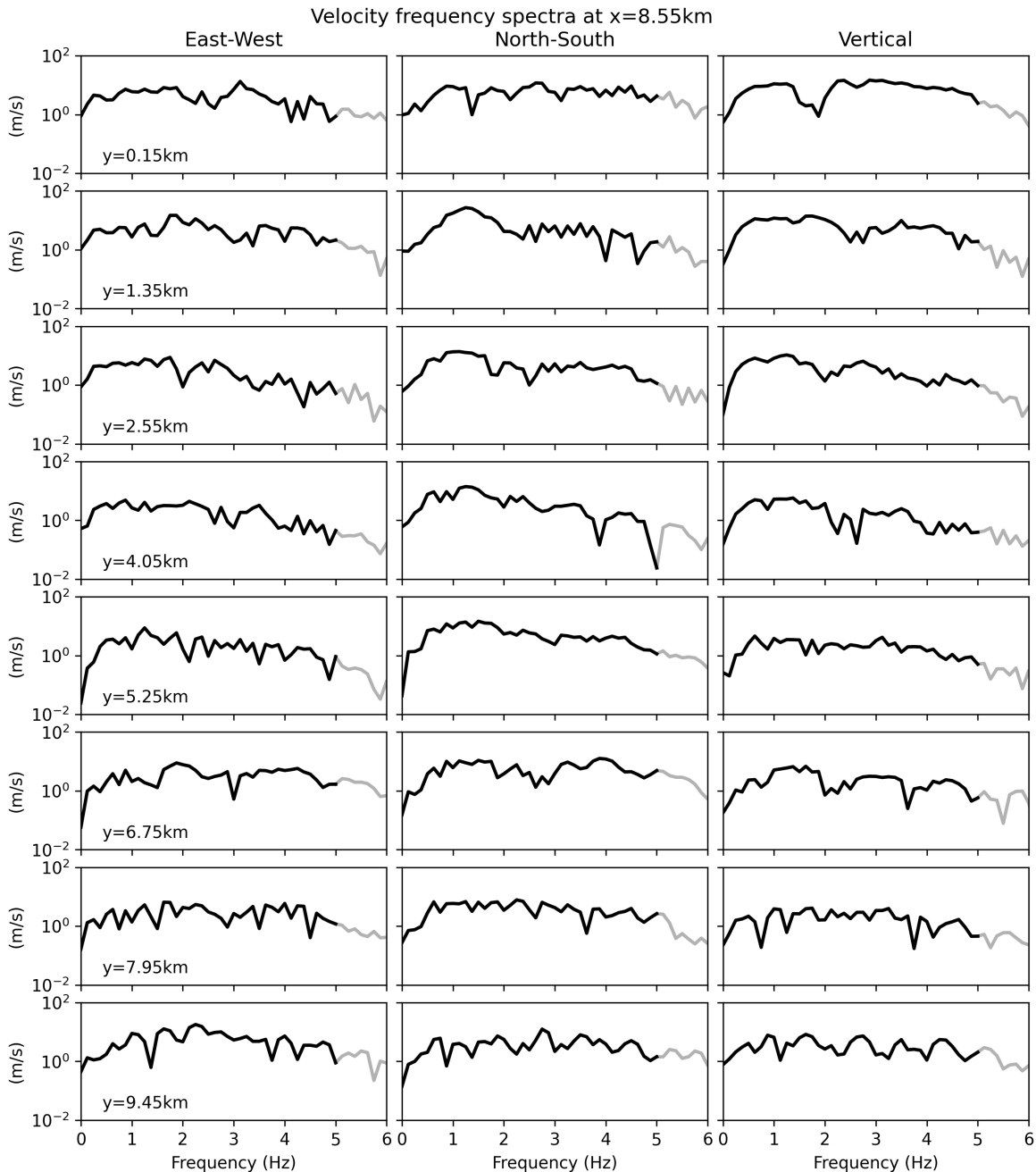

**Figure A3.** Frequency spectra corresponding to Figure 2. Each row corresponds to a different sensor, located on a line parallel to the $y$ axis at $x$=8.85 km. Each column corresponds to one of the three components. The grey line denotes frequencies larger than 5 Hz where numerical simulations are not accurate.

**Appendix B: Dimensionality of data**

## B1 Principal Component Analysis

The intrinsic dimension based on the PCA components has been evaluated with the `scikit-dimension` package. Figure B1 illustrates the number of PCA components required to retain 95% of the data variance depending on the number of samples. It can be noted that, for computational reasons, the velocity fields are represented by a single component (the East-West component, parallel to the $x$ axis) for all three methods.

For comparison purposes, the wavefields intrinsic dimension is also computed for a previous version of the database where the source has a fixed position and orientation (HEMEW-3D database[2]). With this database, the wavefields intrinsic dimension was around 3200. It is reasonable that adding degrees of freedom with a random source increases the variability of data.

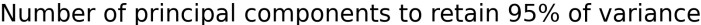

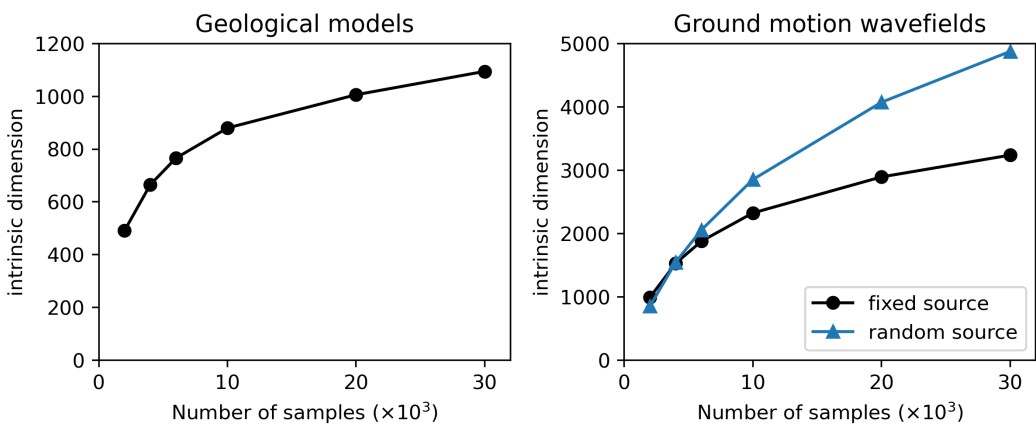

**Figure B1.** Number of principal components ($y$-axis) required to represent 95% of the variance in data as a function of the dataset size ($x$-axis) for geological models (left) and ground motion wavefields (right). For ground motion, the HEMEW-3D database is used for the fixed source (black line) and the HEMEW[S] database corresponds to the blue line.

---

[2]https://entrepot.recherche.data.gouv.fr/dataset.xhtml?persistentId=doi:10.57745/LAI6YU&version=1.0

## B2  Correlation dimension

The correlation dimension is determined as the slope of the linear part in the log-log representation of $C_N$ (Figure B3). This
definition is subject to some interpretation since one should determine which portion constitutes the linear part. Nevertheless,
we found that small variations of the linear part limits had very little influence on the slope estimate (less than one unit).

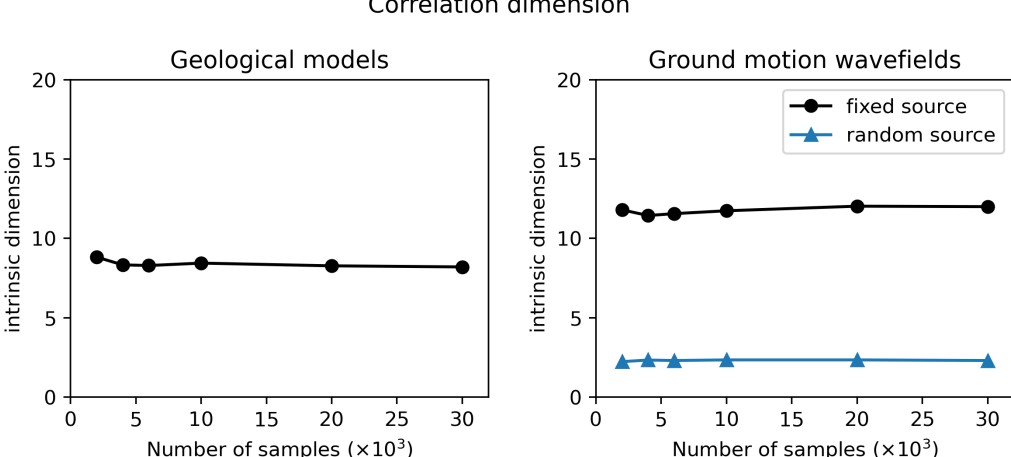

**Figure B2.** Correlation dimension ($y$-axis) as a function of the dataset size ($x$-axis) for geological models (left) and ground motion wavefields
(right). For ground motion, the HEMEW-3D database is used for the fixed source (black line) and the HEMEW$^S$ database corresponds to the
blue line.

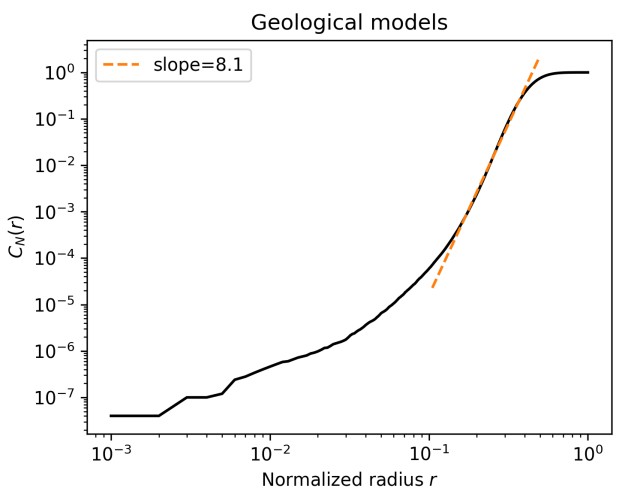

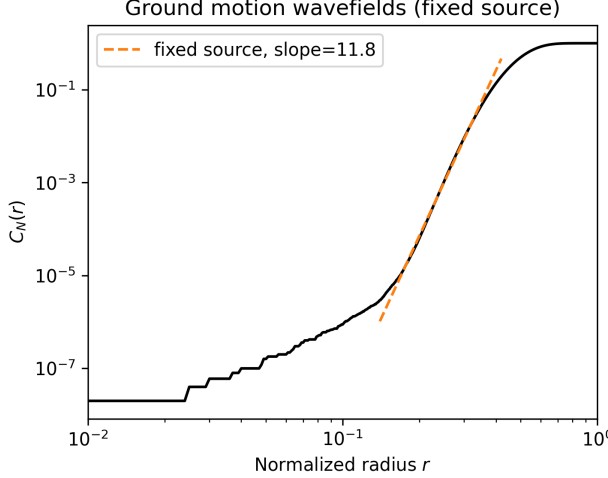

(a) Correlation dimension for $30\,000$ geological models

(b) Correlation dimension for $30\,000$ ground motion wavefields with a fixed source (HEMEW-3D database)

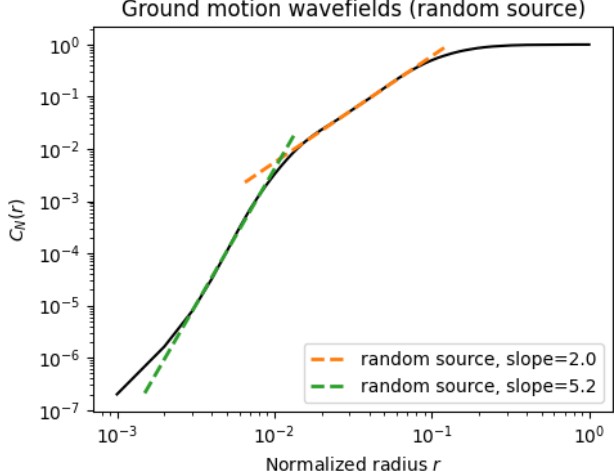

(c) Correlation dimension for $30\,000$ ground motion wavefields with a random source (HEMEW$^\text{S}$ database)

**Figure B3.** The correlation dimension $C_N(r)$ is computed from the number of samples being at (Euclidean) distance smaller than $r$ for different values of $r$ (Equation 8). Then, the correlation dimension is obtained as the slope of the linear part in the log-log representation.

## B3    MLE based intrinsic dimension

The intrinsic dimension based on the Maximum Likelihood Estimator (MLE) has been computed with the `scikit-dimension` package. Figure B4 shows the evolution of the intrinsic dimension as a function of the number of samples for geological models and velocity fields.

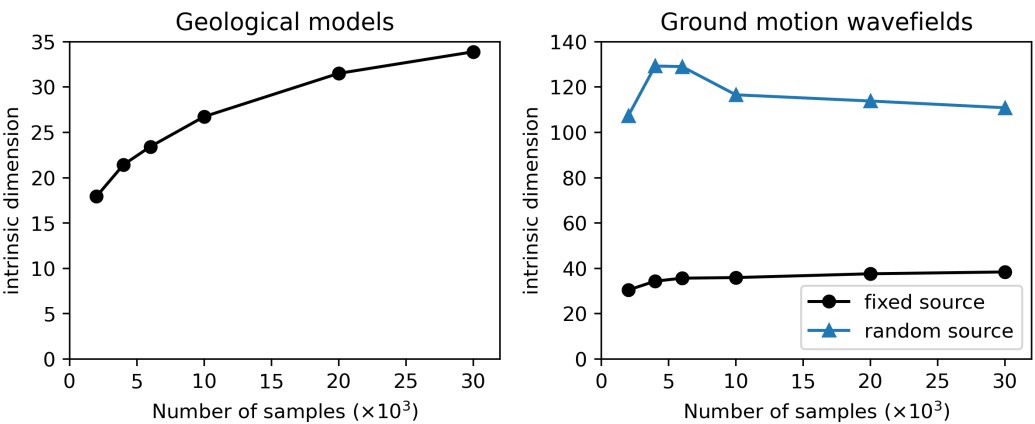

**Figure B4.** Intrinsic dimension estimated by the MLE ($y$-axis) as a function of the dataset size ($x$-axis) for geological models (left) and ground motion wavefields (right). For ground motion, the HEMEW-3D database is used for the fixed source (black line) and the HEMEW[S] database corresponds to the blue line.

## B4 Structural Similarity Index

Figure B5 exemplifies two pairs of geological models with high similarity (SSIM of $0.6$) but different properties. The first pair (Fig. B5a) has similar mean values but different heterogeneities while in the second pair, geological models are almost homogeneous but exhibit different mean values (Fig. B5b).

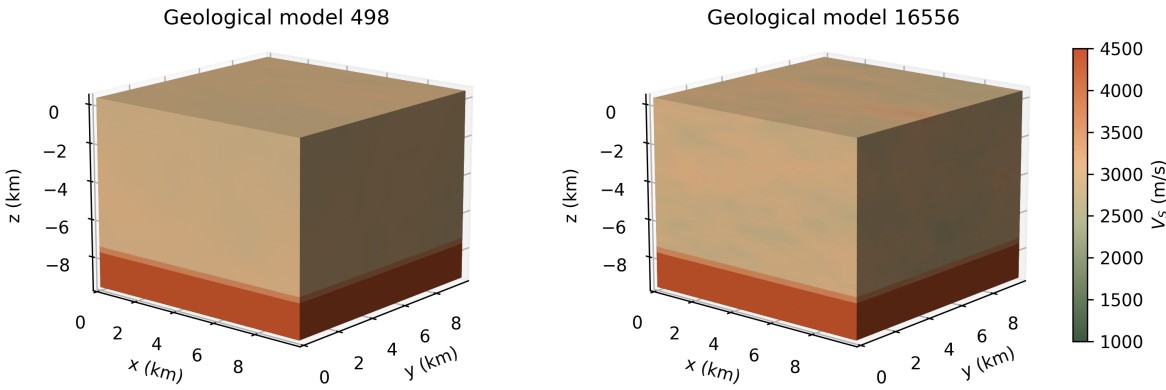

(a) Geological models with SSIM of $0.6$ and normalized distance of $0.03$

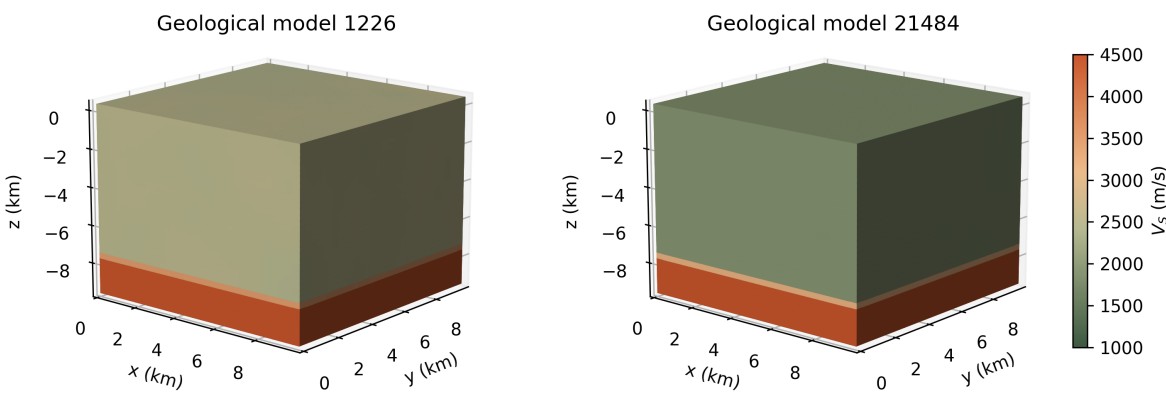

(b) Geological models with SSIM of $0.6$ and normalized distance of $0.15$

**Figure B5.** Two pairs of geological models with a high SSIM of $0.6$

 **Appendix C: Multiple Input Fourier Neural Operator (MIFNO)**

The MIFNO architecture is shown in Fig. C1. Details about the model are given in Lehmann et al. (2024).

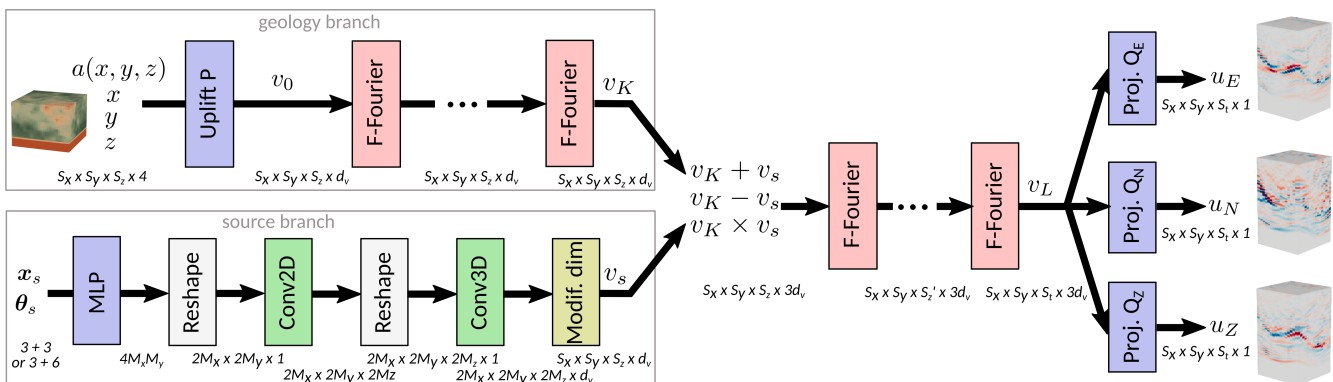

**Figure C1.** The MIFNO is made of a *geology branch* that encodes the geology with factorized Fourier (F-Fourier) layers, and a *source branch* that transforms the vector of source parameters $(\boldsymbol{x}_s, \boldsymbol{\theta}_s)$ into a 4D variable $v_S$ matching the dimensions of the *geology branch* output $v_K$. Outputs of each branch are concatenated after elementary mathematical operations and the remaining factorized Fourier layers are applied. Uplift $P$ and projection $Q_E, Q_N, Q_Z$ blocks are the same as in the F-FNO.

*Author contributions.* F.L., F.G., D.C. designed the study. F.L. conducted the analyses. M.B. and D.C conceived the original idea. F.L. wrote the manuscript with input from all authors.

*Competing interests.* The authors have no competing interest to declare.

*Acknowledgements.* We used the code https://github.com/bakerjw/GMMs/tree/master (last accessed July 9, 2024) to compute the GMMs. The authors are grateful for the resources and human support of the Très Grand Centre de Calcul (TGCC, CCRT, France). They thank the anonymous reviewers for their numerous suggestions that helped to improve the manuscript.

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
