# Peer review of "Synthetic ground motions in heterogeneous geologies from various sources: the HEMEWS-3D database"

_Earth System Science Data, 2023_

## Referee Comment (RC2)

Review of: Synthetic ground motions in heterogeneous geologies: the HEMEW-3D dataset for scientific machine learning

**General:**

This is a well presented, if sometimes a little brief, manuscript describing a dataset of 3d physics-based simulations where the source remains the same (Le Teil earthquake, France 2019) and the deeper 3d geology is randomly varied within a given range of initial conditions, always for rock-to-hard-rock materials. Results (velocity traces) hold up to 5 Hz. It is envisioned that this set be used for enriching the limited empirical sets of seismic recordings. Some thoughts and concerns are expressed in what follows, which do not challenge the procedure or results per se, but rather its interface with earth science and engineering seismology and its usability at large.

**Main/specific comments and concerns:**

There are several references to recent work by the same authors (2022, 2023a, 2023b). It would be very helpful to state with even more detail and clarity the relation, differences and originality of the work at hand with respect to those. Two of these past works are also mentioned in the discussion section as 'applications' (presumably of the current work), which adds to the possible confusion of a reader who may not be already acquainted with this research group's output.

If the main purpose of this dataset is to enrich the existing (limited) recorded datasets for scientific machine learning, then a potential user may expect that its creation should follow strict scientific rigor in terms of simulating the physical phenomena involved, be as comparable as possible to recorded data and actual conditions, and not include any realizations that may be deemed unphysical. This in my view implies that:

- The stratification and overall selection of properties (mostly Vs) in the geological models needs to be plausible, and the randomization needs to be constrained by how formations are found in nature. Enough empirical knowledge exists in this field, which can serve to limit/guide the possible random cases based on credibility. For instance, not only the possibility of Vs reversals but also the impedance contrasts between formations need to be considered from the point of view of geological processes, etc. (contrasts in particular are important, as they determine the amplification levels). Lines 271-275 leave it to the user to perform the 'sanity checks' – on the contrary, if implausible data are left in (which they should not, in my view), they should at least bear clear labels/flags. Moreover, not only the variations/randomizations within the range, but also the range itself needs some justification: e.g., Vs ranges from 1070-4500 m/s for an assumed domain down to 8 km depth. Given that values of 3500 m/s are usually considered appropriate within the crust for even deeper seismogenic depths, is not 4500 m/s rather high?

- The geology is taken into account from the depth where Vs exceeds 1070 m/s and downwards. From the point of view of site response, this means that the top dozens or hundreds of meters of what is usually found as near-surface geological materials is ignored, or in engineering terms, eurocode-8 'A-class' rock sites are assumed. This opens some questions: 1. Rock sites often exhibit amplification at high frequencies (say, >8 Hz): these however would be invisible here, if the simulations only reach up to 5 Hz. Conversely, site response <5 Hz (which could actually be seen in the available bandwidth) is typically related to softer materials (say, Vs<600 m/s) which in turn are not included in the models. So either way, it seems like the site response/amplification is not captured fully, despite the effort to consider so many geology variations. 2. The 'spatial sampling' is of 300 m horizontally. However (and it is often the case also for rock sites with Vs>1000 m/s), the lateral variability can be much stronger, which again would mean that wave phenomena within

distances <300 m would become invisible in these models, though likely important in nature. 3. Are there many regions where a surface Vs>1000 m/s is deemed probable, so that the synthetics here can represent surface motion? If not (as is my belief), is there any recommendation about how these simulations could be coupled with near-surface analyses that would include additional soft material effects (which are what really modifies ground motion in most observed cases) or even topography effects?

- The variability of the synthetic results should be somehow calibrated to that of empirical ground motion data. This means considering the components of what is known as sigma and its components in GMPEs, and comparing to the statistics of the group of simulations. This is mentioned in passing in line 185 (Convertito paper), but I feel it should be addressed more fully. E.g. fig. 3 shows the spread of the data in terms of PGV - how would it compare to observed data? Also, it would be nice to see more analyses and commentary such as that of fig. 3b and lines 186-187 (effect of varying geology on the variability of simulated ground motion) -  this seems to be a rather central point of this exercise not stressed enough. Pages 12-14 focus on capturing variability of the model output from the point of view of number of realizations etc… but I am more concerned about the variability depending on the choice of initial constraints (on Vs, impedance contrast, thicknesses, etc.), which may not be sufficiently well planted into documented reality. This seems to me like a more urgent check to make.

- Making this huge effort only for one specific source (Le Teil seismic event, specific parameters in lines 100+) seems to me to subtract significantly from the usability of this dataset. This event is likely of great interest to France, and such a magnitude is likely of interest to some other stable continental regions, but an entire type of ground motion uncertainty (the between-event variability) is entirely left out of the dataset by keeping to a single magnitude/mechanism/location. It is mentioned as a future prospect/idea to investigate other events, but there also needs to be a reasoning why this study as it stands is self-sufficient and useful for users at large.

How much is PGV (a rather low-frequency parameter) expected to differ from a 'naturally recorded' PGV, given the band limitation of 5 Hz (line 182)? Also, because some disciplines are very strongly interested in acceleration (engineering), i.e. the derivative of the velocity results achieved here, which is much richer in high frequencies, please make some comment as to how viable it would be to derive acceleration time series and PGA values from these simulations, considering the implications of their upper bound of 5 Hz. This is also very important for the computation of Arias intensity (eq. 5), and the reliability of all computed durations hinges on it.
I am concerned about one more thing regarding the significant durations. According to page 11, most simulations have T< 2 or 3 s. However, in the example of page 9 (even if they are velocity traces), the duration by eye seems to be 5 s or more. How is that explained? Also, fig. 4a implies that many synthetics last significantly less than even 1 s, how is that explained? In recorded ground motion datasets, we rarely find such very short durations. Is it an artifact of he 5-Hz limit? If so, please propose a correlation to bring such 'compromised' T values closer to recorded ones.
One last thing that I do not understand: the commentary on fig. 5 says that 'significant ground motion happens between times 1.6-17 s'. But the synthetic only exceeds the P-arrival threshold from 2 s onwards. Even if accurate, I am not sure such a plot combining different time series is so meaningful or useful. Please explain its rationale/necessity.

The 'applications' section mentioned in the discussion could benefit from some more elaborating: 1. if basins are created in the mesh, then their fill material needs to be up to 4-5 times slower than the minimum current Vs - meaning 4-5 times longer run time for the analyses?  2. it is unclear how the 2023a and 2023b publications are applications of the current one; 3. as mentioned before, before synthetics can be used to actually infer conclusions about real ground motion, they should first be calibrated on the natural variability of observations; 4. exploring near-surface features is certainly

needed, as discussed above, but more details as to the how would be welcome here. 0verall, it feels as if some of the issues 'left for later' could perhaps have been somehow included or at least better considered/discussed in this current effort, which –aside from the large number of realizations-seems somewhat limited in scope.

**Other/technical/lesser comments:**

In 2.1, it is unclear how these examples relate to 'data used in geophysics and seismology', except in a rather broad way. This reviewer, and likely the average reader, cannot see the relation in scope between $CO_2$ leakage/flow databases and seismic ground motion simulations. On the other hand, the examples in 2.2, which are more closely related to the topic, could be detailed a little more. It is not fully understandable neither from this paragraph nor from the table exactly how they compare to the effort at hand. Please help by providing more context.

Lines 175+: P arrivals (and other phases) for the time being are most often identified by automated procedures e.g. comparing the short-term to the long-term average (STA, LTA), rather than by analysts or machine learning (though this may change soon). Why would this not be used here, and instead a velocity threshold is used?

The state of the art seems to focus a little more on the region of the authors. Though this is not objectionable, I wonder if it would be possible to make some additional references to the other synthetic works performed in other regions, including e.g. the SCEC broadband simulations.

Fig. 2 shows spatial variability of synthetics for 1 realization. It would be nice to add the 'input motion' (time series at the source) for comparison, and also to show the same figure layout for Fourier amplitude spectra (up t 5 Hz).

Please reconsider and possibly amend the number of decimals appearing in the various quantities described. For instance, page 11: duration of a seismogram cannot conceivably be given at an accuracy of 2 decimals of a second.

Line 160, how is the 100-Hz sampling frequency explained in view of the 5-Hz maximum threshold for wave propagation within the numerical grid?

Line 186: isn't scattering (loss of energy rather than its spreading in time) also a possibility?

A few phrases could be reconsidered in terms of use of language: e.g. words like 'incredibly' or 'tricky' can be avoided.

---

## Author Comment (AC1)

**Synthetic ground motions in heterogeneous geologies from various sources: the HEMEW$^S$-3D database**

Fanny Lehmann[1,2], Filippo Gatti[2], Michaël Bertin[1], and Didier Clouteau[2]

[1]CEA, CEA/DAM/DIF, F-91297 Arpajon, France
[2]LMPS - Laboratoire de Mécanique Paris-Saclay, Université Paris-Saclay, CentraleSupélec, ENS Paris-Saclay, CNRS, Gif-sur-Yvette, France

**Important note on the revised manuscript**

The authors thank the reviewers for their numerous and detailed comments. During the review process, the authors have proposed a new database (HEMEW$^S$-3D ) that addresses several of the shortcomings highlighted by the reviewers. Therefore, the revised manuscript describes this updated database. The main differences between the initial HEMEW-3D and the revised HEMEW$^S$-3D databases are the following:

1. parameters of the geological models (mean value, thickness, coefficient of variation, and correlation lengths in each layer) are now provided as metadata

2. the source has a random position and orientation in HEMEW$^S$-3D

3. the spatial sampling of velocity wavefields was increased from $16 \times 16$ sensors to $32 \times 32$ sensors. The total duration was reduced from $20\,\text{s}$ to $8\,\text{s}$ to maintain reasonable memory requirements and ignore insignificant ground motion.

**1 Response to reviewer 1**

**General comments**

The datasets proposed in this paper consist of (1) 30000 'geological' models and (2) corresponding virtual recordings at 256 surface stations for a double-couple point source excitation at depth. The datasets is certainly unique in the sense that it corresponds to a parametric exploration of a large space with 3D numerical simulation tools in a realistic frequency range and is therefore not the result of a routine process. My opinion is that there are some informations that should be added to favor the reuse of the data. Given that the data are already published on a repository, I do not know whether the curation is possible. In any case, my concerns about data reuse are listed in the paragraph below.

**1.1 Individual scientific questions/issues**

**1.1.1 Generation of the background propagation media: to generate their bank of geological models, the authors consider 7 driving parameters (number of layers, thickness of layers...), the values of which are chosen assuming a uniform probability distribution in predefined intervals. A major problem of this approach is to consider (without explicitly mentioning it) that the parameters are independent. The authors themselves admit that some velocity models may be unrealistic because, for example, the depth dependence of the average velocity is not constrained by any monotonicity conditions, or other constraints like the occurrence of low-velocity layers, the average vertical slowness...**

Our focus is on the physics of wave propagation, which remains valid for all arrangements of layers. The models might be 'unrealistic' for a geologist but they remain physically admissible from a pure wave propagation perspective. The intent is to promote efficient machine learning training by providing rather heterogeneous geological configurations, at multiple scales. From a machine learning perspective, such a variety of profiles better covers the unknown underlying probability distribution, fostering the model's capability of generalization. The independence hypothesis is now clearly stated l.145.

**1.1.2 The same criticism can be done for the correlation lengths used to define the autocovariance function for the small-scale fluctuations: many realistic situations would call for similar horizontal correlation lengths and much smaller (commonly by one order of magnitude) vertical correlation length. The authors argue that non-realistic velocity models can be discarded from the database. This is certainly possible for the average velocity depth dependence which can be measured directly on the geological models, but how would a user proceed to discard unrealistic (or undesired) correlation lengths? It seems indeed that the information about the values of the seven parameters is not present in the geological models. This information should be added as metadata to the geological models to favor the reuse of the synthetic seismograms.**

Metadata about coefficients of variation and correlation lengths have been added in the new version. This allows to discard geological models based on custom criteria. For instance, 4831 geological models have a vertical correlation length of 1500m or 3000m. Again, the models might be 'unrealistic' from a geologist's point of view but they remain physically admissible from a pure wave propagation perspective.

**1.1.3 Another missing information in the article is the number of random draws used for each parameter. Only the total number of models (30000) is given. This number seems large, but how would it reduce once non-realistic models have been discarded?**

For each set of parameters, only one random realization of random fields has been drawn. This has been made explicit in the revised manuscript (l.178). One option to discard unrealistic samples is to remove geological models with low impedance contrasts between successive layers. From the metadata, it can be seen that 10 550 samples out of 30 000 have all impedance contrasts higher than 0.8.

**1.1.4  Generation of the random heterogeneities: the generation of random fluctuations presented in the article poses two potential problems (for the reuse of data sets). First, it seems that only one realization of the random fluctuations is considered. If true (please confirm by explictly writing it), it means that ensemble average would have to be replaced by spatial average (on the 256 virtual sensors) assuming that ergodicity holds. Testing the ergodicity assumption would be quite difficult without the information about the sensor interdistances and some physical lengths that control the scattering regimes. This is in fact the second issue: it is difficult to guess the scattering regime (characterized by the values of the scattering and transport mean free paths) under consideration in the simulations, and this may prevent the reuse of the datasets. It would be useful to add as metadata the mean free path values for each layer. Without this information, the reuse of the datasets would be certainly limited.**

Each profile within the dataset is an independent realization of a 3D non-stationary random field. For each sample, one realization of the random fluctuation is sampled and added to the varying average. The sensor inter-distance is fixed to 300 m. Guessing the scattering regime is out of scope of this publication, which tailors machine learning applications, which are not prevented by the lack of this information. Finally, ergodicity is implicitly provided by the properties of the random field generator, called `randomField` (refer to [de Carvalho Paludo et al., 2019] for further details).

**1.2  Technical corrections**

**1.2.1  The choice of the sampling parameters (in space and time) for the synthetic datasets is rather surprising: on one hand, the time series are sampled at 100 Hz frequency, whereas the maximum simulated frequency is 5 Hz. This corresponds to a 10-fold additional cost in terms of storage.**

100 Hz matches the usual temporal resolution of recorded time series available in public accessible earthquake engineering strong motion databases which is important for tasks such as seismic phase picking.

**1.2.2  On the other hand, the spatial sampling is limited to the minimum possible wavelength (300 m), whereas it should be twice finer (assuming that a wavelength will not be sensitive to heterogeneities smaller than half its size).**

The spatial sampling of geological models is given with the minimal resolution that allows to generate accurate wavefields. However, a higher spatial resolution was used in the internal routines to compute random fields and velocity wavefields. In particular, the smallest correlation length (i.e. 1500 m) equals five times the minimum wavelength.

**1.2.3  The content of the README.md file seems to be uncorrect. It states that "The 300 velocity fields files amount to 2.96Tb. They are downloadable individually (9.8 Gb per file)", but each velocity file weights only (and fortunately) 1.1 Gb. Please explain and correct.**

The misunderstanding comes from a misuse of bits instead of bytes. This has been corrected in the new readme file which now indicates 263.4 GB.

**1.2.4  Give the value of N in section 3.3.2**

N depends on the choice of the power spectral density (PSD), which is the von Karman correlation in the current work. The `randomField` code adopted to generate random fields is implemented in such a way that i) the PSD is well represented and ii) the discretization of the PSD is sufficiently fine to introduce enough randomness in the domain under consideration. About the former, the value of $N$ ensures that the integral of the approximate PSD amounts to at least $95\%$ of the true one. About the latter, the choice of $\Delta k$ is small with respect to the inverse of the correlation length.

**2 Response to reviewer 2**

**General**

This is a well presented, if sometimes a little brief, manuscript describing a dataset of 3d physics-based simulations where the source remains the same (Le Teil earthquake, France 2019) and the deeper 3d geology is randomly varied within a given range of initial conditions, always for rock-to-hard-rock materials. Results (velocity traces) hold up to 5 Hz. It is envisioned that this set be used for enriching the limited empirical sets of seismic recordings. Some thoughts and concerns are expressed in what follows, which do not challenge the procedure or results per se, but rather its interface with earth science and engineering seismology and its usability at large.

**2.1 Main/specific comments and concerns**

**2.1.1 There are several references to recent work by the same authors (2022, 2023a, 2023b). It would be very helpful to state with even more detail and clarity the relation, differences and originality of the work at hand with respect to those. Two of these past works are also mentioned in the discussion section as 'applications' (presumably of the current work), which adds to the possible confusion of a reader who may not be already acquainted with this research group's output.**

The 'applications' section 5 has been rewritten to provide more background on the previous works while keeping the scope of the current paper on the database description. All mentioned works use the HEMEW$^S$-3D database as a starting point for further applications but do not dive into the details of the database characteristics.

**If the main purpose of this dataset is to enrich the existing (limited) recorded datasets for scientific machine learning, then a potential user may expect that its creation should follow strict scientific rigor in terms of simulating the physical phenomena involved, be as comparable as possible to recorded data and actual conditions, and not include any realizations that may be deemed unphysical. This in my view implies that:**

**2.1.2 The stratification and overall selection of properties (mostly Vs) in the geological models needs to be plausible, and the randomization needs to be constrained by how formations are found in nature.**

The strict scientific rigor that the reviewer claims has been definitely respected, by not only considering the geological aspects (the values of the shear-wave velocity are taken within reasonable limits) but also by spanning the widest variety of geological profile possible, including those that we would call 'extremely rare' geological profiles. This strategy has been designed in order to reduce the learning bias that any machine learning algorithm, solely trained on standard geological configurations, would show. Introducing 'extremely rare' yet physically admissible geology samples strengthens the learning process, fostering generalizability and enhancing the representation power of the machine learning algorithms. As a matter of fact, the presence of 'extremely rare' geological profiles implies that the coda of the associated unknown probability distribution are sampled. Other geological profiles conceived for machine learning purposes adopt the same sampling strategy (openFWI).

**2.1.3 Enough empirical knowledge exists in this field, which can serve to limit/guide the possible random cases based on credibility. For instance, not only the possibility of Vs reversals but also the impedance contrasts between formations need to be considered from the point of view of geological processes, etc. (contrasts in particular are important, as they determine the amplification levels).**

The credibility of the generalization process must be ensured not only from a pure geological perspective but also from a statistical point of view. Therefore, as mentioned in the previous answer, we opted for introducing 'extremely rare' samples, that foster the overall learning process. Moreover, one option to discard unrealistic samples is to remove geological models with low impedance contrasts between successive layers (from the provided metadata). For instance, 10 550 samples out of 30 000 have all impedance contrasts higher than 0.8.

**2.1.4 Lines 271-275 leave it to the user to perform the 'sanity checks' − on the contrary, if implausible data are left in (which they should not, in my view), they should at least bear clear labels/flags.**

Metadata have been added in the new version to enable filtering. We do believe that implausible data depend on the targeted application.

**2.1.5    Moreover, not only the variations/randomizations within the range, but also the range itself needs some justification: e.g., Vs ranges from 1070-4500 m/s for an assumed domain down to 8 km depth. Given that values of 3500 m/s are usually considered appropriate within the crust for even deeper seismogenic depths, is not 4500 m/s rather high?**

Those values are in line with the EPcrust model [Molinari and Morelli, 2011], which justifies their choice. Hard-rock sites in Japan, for instance, YMGH01 KiK-net site, reach a shear-wave velocity of 3000 m/s at 200 m depth (according Figure 6 [Nakano et al., 2015]), which leads to think that a shear-wave velocity value of 4500 m/s at 8 km depth is reasonable.

**The geology is taken into account from the depth where Vs exceeds 1070 m/s and downwards. From the point of view of site response, this means that the top dozens or hundreds of meters of what is usually found as near-surface geological materials is ignored, or in engineering terms, eurocode-8 'A-class' rock sites are assumed. This opens some questions:**

**2.1.6    Rock sites often exhibit amplification at high frequencies (say, >8 Hz): these however would be invisible here, if the simulations only reach up to 5 Hz. Conversely, site response <5 Hz (which could actually be seen in the available bandwidth) is typically related to softer materials (say, Vs<600 m/s) which in turn are not included in the models. So either way, it seems like the site response/amplification is not captured fully, despite the effort to consider so many geology variations.**

The choice of the shear-wave velocity values is the result of a tradeoff between implied computational cost and realism of the rendered regional ground motion. As an example, [Sochala et al., 2020] performed 400 large-scale simulations over randomly fluctuating geologies, which are insufficient for machine learning purposes, even more when considering the overall 4 million core-hours CPU time required to render those results. Our final target is to provide scientists with a regional ground motion dataset to develop robust machine learning surrogates, mostly focusing on the role of multi-scale heterogeneity, which represents the real difficulty to tackle when attempting to surrogate the 3D wave equation. Moreover, site-effects are mostly controlled by the values of the impedance factors, which are widely covered by this database. A detailed representation of the full range of site-effects would definitely require finer meshes, in order to discretize shorter wavelengths and thereby reproducing site-effects at higher frequencies – assuming constant shear-wave velocity lower bound - or, alternatively, introduce new samples featured by lower shear-wave velocity values. Moreover, site-effects taking place in softer soil deposits are often associated to some sort of mild-to-large non-linearity, at a geotechnical level of investigation. This aspect is out of the scope of this paper. Finally, one should mention that there are not that many contributions in the literature that reached such frequency limit (and implied computational burden) over 30000 simulations (as an example, [Sochala et al., 2020] considered only 400 simulations with maximum accuracy of 3 Hz. Despite covering a range of shear-wave velocity values from 104 to 2100 m/s, they focused on 1D random profiles in the sedimentary layer).

**2.1.7    The 'spatial sampling' is of 300 m horizontally. However (and it is often the case also for rock sites with Vs>1000 m/s), the lateral variability can be much stronger, which again would mean that wave phenomena within distances <300 m would become invisible in these models, though likely important in nature.**

If the reviewer refers to the spatial sampling of the geological models, it follows common practice in numerical simulation where the spatial step should be at most equal to the smallest wavelength. Concerning the spatial sampling of velocity fields, it already allows a good representation of wave propagation. Knowing that the velocity fields database already amounts to 263 GB, increasing the spatial sampling would increase the memory requirements and limit the usability with common devices.

**2.1.8    Are there many regions where a surface Vs>1000 m/s is deemed probable, so that the synthetics here can represent surface motion? If not (as is my belief), is there any recommendation about how these simulations could be coupled with near-surface analyses that would include additional soft material effects (which are what really modifies ground motion in most observed cases) or even topography effects?**

As is now explicitly stated in the main text (l.350-351), surface velocity is understood as the mean velocity in the first 300 m and cannot be compared with, for instance, $V_{S,30}$. As already mentioned, regional velocity models commonly show surface velocity larger than 1000 m/s (e.g. [Causse et al., 2021]). Even if we were to compare with $V_{S,30}$, [Ramadan et al., 2024] showed that 38 stations out of 121 in France have a $V_{S,30}$ between 800 m/s and 1500 m/s and 20 have a $V_{S,30}$ larger than 1500 m/s. Therefore, our range of $V_S$ values is reasonable in different regions.

Simulations could eventually be coupled with refined near-field models by standard convolution techniques. Including topography would require new simulations, but this is out of scope of this paper, which still represents the very first attempt at rendering such a large synthetic database of 3D wave propagation problems in highly heterogeneous media.

**2.1.9 The variability of the synthetic results should be somehow calibrated to that of empirical ground motion data. This means considering the components of what is known as sigma and its components in GMPEs, and comparing to the statistics of the group of simulations. This is mentioned in passing in line 185 (Convertito paper), but I feel it should be addressed more fully. E.g. fig. 3 shows the spread of the data in terms of PGV - how would it compare to observed data?**

There is no interest in comparing with GMPE because the database was neither tailored on a specific seismotectonic framework, nor focused on a specific region of the world. Any of these two constraints would have biased the database and therefore hampered its use for training any machine learning algorithm. Even if we sketched a comparison, we should have used generic GMPEs, compatible with the choice of focusing neither on specific regions nor on specific seismotectonic contexts. Unfortunately, most of the GMPEs of generic use are rarely robustly calibrated at short distance, i.e. within the 10 km distance to the hypocenter, which represents the setup of our simulations. Therefore, GMPEs would probably be poorly reliable for comparison purposes. We instead attempted at reproducing the widest range of cases and at spanning the unknown probability distribution, yet preserving the physics of the wave propagation phenomenon, ensured by the validated and verified wave propagation solver SEM3D. GMPEs are empirically calibrated and very rarely (or never) take into account complex wave propagation phenomena, such as multi-path scattering through randomly fluctuating multi-scale heterogeneity.

**2.1.10 Also, it would be nice to see more analyses and commentary such as that of fig. 3b and lines 186-187 (effect of varying geology on the variability of simulated ground motion) - this seems to be a rather central point of this exercise not stressed enough.**

More analyses have been added in the revised manuscript.

**2.1.11 Pages 12-14 focus on capturing variability of the model output from the point of view of number of realizations etc... but I am more concerned about the variability depending on the choice of initial constraints (on Vs, impedance contrast, thicknesses, etc.), which may not be sufficiently well planted into documented reality. This seems to me like a more urgent check to make.**

Updated analyses in Section 3 provide additional results on the variability of intensity measures. For instance, the PGV distributions in Fig. 5 show that the PGV spans three orders of magnitude. The influence of epicentral distance, mean velocity, and velocity at the source location is also quantified on several intensity measures.

**2.1.12 Making this huge effort only for one specific source (Le Teil seismic event, specific parameters in lines 100+) seems to me to subtract significantly from the usability of this dataset. This event is likely of great interest to France, and such a magnitude is likely of interest to some other stable continental regions, but an entire type of ground motion uncertainty (the between-event variability) is entirely left out of the dataset by keeping to a single magnitude/mechanism/location. It is mentioned as a future prospect/idea to investigate other events, but there also needs to be a reasoning why this study as it stands is self-sufficient and useful for users at large.**

The new database adds variability in the source location and orientation.

**2.1.13 How much is PGV (a rather low-frequency parameter) expected to differ from a 'naturally recorded' PGV, given the band limitation of 5 Hz (line 182)?**

This remark is not central in the current work as our results can only be compared with records in the validity range of numerical simulations. It is meant to remind readers that may be unfamiliar with numerical settings that some limitations exist. Quantitatively on a small database at our disposition, we obtained a mean PGV underestimation of $-4\%$ and up to $-40\%$ in rare cases.

**2.1.14** **Also, because some disciplines are very strongly interested in acceleration (engineering), i.e. the derivative of the velocity results achieved here, which is much richer in high frequencies, please make some comment as to how viable it would be to derive acceleration time series and PGA values from these simulations, considering the implications of their upper bound of 5 Hz. This is also very important for the computation of Arias intensity (eq. 5), and the reliability of all computed durations hinges on it.**

Thanks to the time temporal sampling (100 Hz), acceleration can be computed as the derivative of velocity time series. However, frequencies larger than 5 Hz cannot be obtained. Although we acknowledge the importance of high-frequency acceleration for engineering applications, they are not obtained from numerical simulations, even for the most demanding high-frequency simulations up 15 Hz.

**2.1.15** **I am concerned about one more thing regarding the significant durations. According to page 11, most simulations have T< 2 or 3 s. However, in the example of page 9 (even if they are velocity traces), the duration by eye seems to be 5 s or more. How is that explained? Also, fig. 4a implies that many synthetics last significantly less than even 1 s, how is that explained? In recorded ground motion datasets, we rarely find such very short durations. Is it an artifact of the 5-Hz limit? If so, please propose a correlation to bring such 'compromised' T values closer to recorded ones.**

The revised manuscript illustrates the relative significant duration (RSD) (Fig. 2) for several time series. The short RSDs are indeed related to the absence of high-frequency components in the coda and the dominance of high pulse-like motions in some cases.

**2.1.16** **One last thing that I do not understand: the commentary on fig. 5 says that 'significant ground motion happens between times 1.6-17 s'. But the synthetic only exceeds the P-arrival threshold from 2 s onwards. Even if accurate, I am not sure such a plot combining different time series is so meaningful or useful. Please explain its rationale/necessity.**

This figure was misleading with the log scale and it has been removed in the revised manuscript.

**The 'applications' section mentioned in the discussion could benefit from some more elaborating:**

**2.1.17** **if basins are created in the mesh, then their fill material needs to be up to 4-5 times slower than the minimum current Vs - meaning 4-5 times longer run time for the analyses?**

A minimum velocity values 5 times slower than the current one implies $5^3$ more elements and therefore, approximately $5^3$ longer run times.

**2.1.18** **it is unclear how the 2023a and 2023b publications are applications of the current one**

The applications section has been clarified.

**2.1.19** **as mentioned before, before synthetics can be used to actually infer conclusions about real ground motion, they should first be calibrated on the natural variability of observations. Exploring near-surface features is certainly needed, as discussed above, but more details as to the how would be welcome here. Overall, it feels as if some of the issues 'left for later' could perhaps have been somehow included or at least better considered/discussed in this current effort, which –aside from the large number of realizations-seems somewhat limited in scope**

The database we propose in targeted towards machine learning applications and it is out of scope to investigate numerous seismological applications.

**2.2 Other/technical/lesser comments**

**2.2.1** **In 2.1, it is unclear how these examples relate to 'data used in geophysics and seismology', except in a rather broad way. This reviewer, and likely the average reader, cannot see the relation in scope between CO2 leakage/flow databases and seismic ground motion simulations.**

This section has been clarified to highlight the mathematical similarities between both frameworks which is related to the hyperbolicity of the governing equations (l.68-71).

**2.2.2** On the other hand, the examples in 2.2, which are more closely related to the topic, could be detailed a little more. It is not fully understandable neither from this paragraph nor from the table exactly how they compare to the effort at hand. Please help by providing more context.

This section has been revised.

**2.2.3** Lines 175+: P arrivals (and other phases) for the time being are most often identified by automated procedures e.g. comparing the short-term to the long-term average (STA, LTA), rather than by analysts or machine learning (though this may change soon). Why would this not be used here, and instead a velocity threshold is used?

The mentioned procedures have been developed for recordings and it is not obvious to apply them on numerical results. Since numerical ground motion is not influenced by any ambiant noise, time series are almost zero before the first wave arrival. Therefore, P arrivals can be obtained easily and accurately with a threshold.

**2.2.4** The state of the art seems to focus a little more on the region of the authors. Though this is not objectionable, I wonder if it would be possible to make some additional references to the other synthetic works performed in other regions, including e.g. the SCEC broadband simulations.

**2.2.5** Fig. 2 shows spatial variability of synthetics for 1 realization. It would be nice to add the 'input motion' (time series at the source) for comparison, and also to show the same figure layout for Fourier amplitude spectra (up to 5 Hz).

Illustration of the source time function has been added as Fig. A1 and Fourier amplitude spectra are shown in Fig. A2.

**2.2.6** Please reconsider and possibly amend the number of decimals appearing in the various quantities described. For instance, page 11: duration of a seismogram cannot conceivably be given at an accuracy of 2 decimals of a second.

Results are now given with a single decimal.

**2.2.7** Line 160, how is the 100-Hz sampling frequency explained in view of the 5-Hz maximum threshold for wave propagation within the numerical grid?

100 Hz matches the usual temporal resolution of recorded time series available in public accessible earthquake engineering strong motion databases which is important for tasks such as seismic phase picking.

**2.2.8** Line 186: isn't scattering (loss of energy rather than its spreading in time) also a possibility?

This figure has been removed from the revised manuscript due to a larger influence of source variability that hides the influence of geological heterogeneities.

**2.2.9** A few phrases could be reconsidered in terms of use of language: e.g. words like 'incredibly' or 'tricky' can be avoided

These sentences have been rephrased.

**References**

[Causse et al., 2021] Causse, M., Cornou, C., Maufroy, E., Grasso, J.-R., Baillet, L., and El Haber, E. (2021). Exceptional ground motion during the shallow Mw 4.9 2019 Le Teil earthquake, France. *Communications Earth & Environment*, 2(1):14.

[de Carvalho Paludo et al., 2019] de Carvalho Paludo, L., Bouvier, V., and Cottereau, R. (2019). Scalable parallel scheme for sampling of Gaussian random fields over very large domains: Parallel scheme for sampling of random fields over very large domains. *International Journal for Numerical Methods in Engineering*, 117(8):845–859.

[Molinari and Morelli, 2011] Molinari, I. and Morelli, A. (2011). EPcrust: A reference crustal model for the European Plate: EPcrust. *Geophysical Journal International*, 185(1):352–364.

[Nakano et al., 2015] Nakano, K., Matsushima, S., and Kawase, H. (2015). Statistical properties of strong ground motions from the generalized spectral inversion of data observed by K-NET, KiK-net, and the JMA Shindokei network in Japan. *Bulletin of the Seismological Society of America*, 105(5):2662–2680.

[Ramadan et al., 2024] Ramadan, F., Lanzano, G., Pacor, F., Felicetta, C., Smerzini, C., and Traversa, P. (2024). Adjusting an active shallow crustal ground motion model to regions with scarce data: Application to France. *Bulletin of Earthquake Engineering*.

[Sochala et al., 2020] Sochala, P., De Martin, F., and Le Maître, O. (2020). Model reduction for large-scale earthquake simulation in an uncertain 3D medium. *International Journal for Uncertainty Quantification*, 10(2):101–127.

---

## Author Response (AR2)

**Synthetic ground motions in heterogeneous geologies from various sources: the HEMEW$^S$-3D database**

Fanny Lehmann[1,2], Filippo Gatti[2], Michaël Bertin[1], and Didier Clouteau[2]

[1]CEA, CEA/DAM/DIF, F-91297 Arpajon, France
[2]LMPS - Laboratoire de Mécanique Paris-Saclay, Université Paris-Saclay, CentraleSupélec, ENS Paris-Saclay, CNRS, Gif-sur-Yvette, France

**Suggestions for revision**

**The authors have added some metadata labels to the database which were indeed needed, and have revised certain parts of the manuscript and its presentation. However, concerning the point-to-point rebuttal, although the authors have written many replies to the reviewer comments, the majority of them are not reflected in the manuscript. It would be a simple thing to add some of the explanations given to the reviewers into the main text, so that they are available to the general readership in a straightforward way, helping towards a better understanding –and most importantly, a better use of the data on offer. So my main and final recommendation for revision is to add explanations given in the rebuttal (and references) to help clarify/improve the main text. Some examples follow:**

The authors thank the reviewer for their detailed comments and their accompanying explanation. Details have been added to the manuscript and a point-by-point response is given below.

**1. Points 2.2.7, 1.2.1, 2.1.14: Both reviewers pose the question of how 5 Hz (maximum frequency that can numerically propagate through a grid) and 100 Hz (sampling rate) are really reconciled. The (identical) reply given to both reviewers is this: "100 Hz matches the usual temporal resolution of recorded time series available in public accessible earthquake engineering strong motion databases which is important for tasks such as seismic phase picking". Yet this is not explained in the revised manuscript, but only given as a personal reply in the rebuttal. But then the reader, who will likely ask him/herself the same, cannot benefit in the end. He/she should not have to read the commentary exchange in order to get the necessary clarifications for the article, so please explain your rationale in the paper.**

**A note regarding this specific reply: Please rephrase this explanation before adding it to the manuscript, because it is incorrect on a few accounts:**

**1. earthquake engineers do not access strong motion datasets to do phase picking, which is a purely seismological task/skill**

**2. the reason for the high sampling in strong-motion data is not for the sake of phase picking (wave windowing can be very rough in such applications, in stark contrast to seismic monitoring) – this investment is made in order to be sure to catch PGA correctly**

**3. in many important networks, the sampling rate of accelerometric data is actually not even 100 Hz but 200 Hz**

Explanation for the choice of 100 Hz have been given l. 210-214, taking into account the reviewer's suggestions. It now reads "Although the sampling frequency is higher than the Nyquist frequency (i.e. $2 \times f_{max} = 10\,\text{Hz}$), the value of 100 Hz was chosen to match the temporal resolution of recorded time series in several publicly accessible datasets (e.g. STEAD [Mousavi et al., 2019]), INSTANCE [Michelini et al., 2021]). The sampling frequency is sufficient to allow an accurate computation of Peak Ground Velocity (PGV), derive the acceleration time series with finite differences, and compute the Peak Ground Acceleration (PGA). "

**2. Points 2.1.2, 2.1.3, 1.1.1, 1.1.2, 2.1.5, 2.1.19: Again both reviewers pointed this out: the choice to include unphysical instances of various parameter values in the database. If the authors agree that some of the models are unrealistic from a geological/geophysical/seismological point of view, then please stress this in the text, and explain why you think there is this dire necessity to include them nevertheless for ML purposes. Also, and this is something I'd like to stress, please be very clear on what percentage of the data can be related to unrealistic, or in statistical terms, "extremely rare" or "coda" cases. Because it seems as if these rare cases may actually take up a lot of the database: from the numbers the authors give, it seems like a ratio of 1:3 between rare/unrealistic and normal (10,550 out of 30,000?), which seems too high, so is the coda being sampled or oversampled in the end? Please be clear on the statistics.**

**To say that the 'plausibility of data depends on the application' is, I think, a compromise detrimental to the earth science applicability of the work, in favor of ML. But natural occurrence does need to have a role here. (E.g. one may well sample 1,000,000 soil samples but will never get a density of, say, $5t/m^3$, and even so, it would certainly not happen 30% of the time.) So please add explicit commentary to the paper about all this. If 2/2 reviewers felt the need to bring it up, most earth scientists in the audience are likely to have similar questions.**

Apart from agreeing that a density of $5\,t/m^3$ is truly unrealistic and stating that no such case is included in the database, it is believed that neither the authors nor the reviewers could give an objective measure of what is a "realistic" model and thus, be able to compute the percentage of such cases in the HEMEW$^S$-3D database. If we were to take impedance contrast as as criterium to discriminate between two types of geologies, the proposed database contains less than $1/3$ of geologies with a minimum impedance contrast lower than 0.7 (l. 403-404). However, from a machine learning perspective, it is extremely important to provide *out-of-distribution* examples in order to demonstrate the generalization capabilities of proposed methods. Hence, our database offers users to select their own measure of *in-distribution* geologies on which their model will be trained.

**3. Point 2.1.6: amplification. I find the arguments of the authors incomplete with respect to the existing knowledge on site response. Even so, please give your arguments in the paper explaining if, how and why amplification is or is not accounted for in your calculations, especially with respect to the frequency range <5 Hz (which may well be impressively high for such calculations but is lower than the range where hard sites amplify), and especially considering the large proportion of high-Vs cases. It is ok to say that it is not accounted for completely, but is at least dealt with better than it was in past papers, or is out of scope, etc. However, I think it is not ok to claim that there is absolutely nothing to talk about here and just ignore the issue: limitations exist and should be stated.**

Ground motion amplification and site effects due to soft sediments are not the targeted applications of the HEMEW$^S$-3D database. An improper formulation of the section 5.3. on "Other potential applications" led to confusion on this point. This section has been rephrased. In addition, the limitations section 6. now clearly mentions that the HEMEW$^S$-3D database is not suited for site effects related to sedimentary basins (l. 384-385).

**4. Limitations related to calculation speed/run time and memory/storage needs seem to underpin many of the decisions made and/or are the answer to many of the reviewer comments (duration of waveforms, spatial sampling, inability to fill basins with soft material, etc). It would be good to explain all these cases together in the end of the paper. So to speak, answer the question: when computing becomes faster/easier in future, what would be the top 5 things you'd like to do differently, without the need to worry about such issues?**

As HEMEW$^S$-3D is the first large-scale 3D database of seismological simulations for machine learning, it was crucial to ensure that the simulation objectives were reachable and that their outcomes remained manageable, both in terms of the memory constraints to store/download/reuse data and in terms of data variability that could make machine learning tasks too complex to be learnt. This is why feasibility concerns dictated several choices. Now that many predictive tasks have been proven possible, one can envision meaningful ways to extend the database. Following the reviewer's suggestion, perspectives are now listed l. 410-417.

**5. points 2.1.8: I did not find this new explanation in the new text about surface velocity. Please add if missing, because it is very important the reader understands how you define 'surface velocity (30 or 300 m!). In the majority of locations in the world where seismic hazard is a concern, we would love to have 'near-surface' Vs of 1000m/s or more, but don't! Unless the dataset is more representative of certain regions (France? stable continental areas?), which it claims it is not. Also, please state in main text (as perspectives – we know it is not in the scope of this paper) if/how your methods and data could be combined with near-surface site effects calculations.**

Explanation about the understanding of surface velocity was given at the beginning of the Limitations section (now l.385-389 in the revised manuscript). It is now referred to as upper velocity instead of surface velocity to avoid confusion. The definition of the upper velocity comes from the vertical resolution of **regional** geological models, which is 300 m. This means that Gauss points between 0 m and $-300$ m have the same value. As a consequence, the upper velocity should be understood as a 300-m average and cannot be compared with more common definitions, such as $V_{S,30}$.
Concerning the applications for near-surface site effects, the surface ground motion can be considered as an "outcropping bedrock" response which is classically used in 1D site-effect and Soil-Structure Interaction analyses, and which may require deconvolution (l. 376-377).

**2.1.9: GMPEs. Although many GMPEs exist that are informed by simulations, even assuming they were all empirical, they still are a key tool in practice. And so if your database were to show a great divergence from what they predict, it would be extremely important to point it out. A comparison would be beneficial, and if there is disagreement then the various arguments the authors give can be proposed to explain why their work is better fitted than GMPEs for such and such a case. It is not a matter of believing in data more than in simulations, but there is an urgent need that the two communities finally start to acknowledge each other for science to move forward faster. Please help in this direction.**

Comparisons with four GMPEs by [Atkinson, 2015], [Atkinson and Boore, 2006], [Chiou and Youngs, 2014], and [Shahjouei and Pezeshk, 2016] have been added in Fig. 8. They show a good agreement between the PSA computed from the HEMEW$^S$-3D database and the PSA from GMPEs. Figure 8 is reproduced below

[Figure]

Figure 1: PSA at period T=0.2 s as a function of hypocentral distance (left) and epicentral distance (right) for GMMs by [Atkinson, 2015] (purple, left), [Atkinson and Boore, 2006] (orange, right), [Chiou and Youngs, 2014] (blue, right), [Shahjouei and Pezeshk, 2016] (green, right), and our HEMEW$^S$-3D database. Solid lines correspond to the mean PSA and shaded areas to one standard deviation.

**2.1.15: please make this clarification in text about durations and lack of content**

Details have been added l. 250-251 and now read "These short RSD values are related to the absence of high-frequency components in the coda and the dominance of high pulse-like time series in cases with shallow sources and low heterogeneity contrasts."

**2.2.4: this was not answered**

A dedicated paragraph 2.1. has been added in the Related work Section.

**Outside reviewer bullet points: On lowering duration from 20 sec to 8 sec: Please include a phrase about why 8 sec only is sufficient duration (maybe based on distance and M combinations)**

The choice of 8s has been justified l.251-252 from the consideration of P-wave arrival time and Relative Significant Duration that gives an estimate of the final time of significant ground motion.

**References**

[Atkinson, 2015] Atkinson, G. M. (2015). Ground-Motion Prediction Equation for Small-to-Moderate Events at Short Hypocentral Distances, with Application to Induced-Seismicity Hazards. *Bulletin of the Seismological Society of America*, 105(2A):981–992.

[Atkinson and Boore, 2006] Atkinson, G. M. and Boore, D. M. (2006). Earthquake Ground-Motion Prediction Equations for Eastern North America. *Bulletin of the Seismological Society of America*, 96(6):2181–2205.

[Chiou and Youngs, 2014] Chiou, B. S.-J. and Youngs, R. R. (2014). Update of the Chiou and Youngs NGA Model for the Average Horizontal Component of Peak Ground Motion and Response Spectra. *Earthquake Spectra*, 30(3):1117–1153.

[Michelini et al., 2021] Michelini, A., Cianetti, S., Gaviano, S., Giunchi, C., Jozinović, D., and Lauciani, V. (2021). INSTANCE – the Italian seismic dataset for machine learning. *Earth System Science Data*, 13(12):5509–5544.

[Mousavi et al., 2019] Mousavi, S. M., Sheng, Y., Zhu, W., and Beroza, G. C. (2019). STanford EArthquake Dataset (STEAD): A Global Data Set of Seismic Signals for AI. *IEEE Access*, 7.

[Shahjouei and Pezeshk, 2016] Shahjouei, A. and Pezeshk, S. (2016). Alternative Hybrid Empirical Ground-Motion Model for Central and Eastern North America Using Hybrid Simulations and NGA-West2 Models. *Bulletin of the Seismological Society of America*, 106(2):734–754.